# FUNCTION SPACES WITHOUT KERNELS: LEARNING COMPACT HILBERT SPACE REPRESENTATIONS

**Su Ann Low**
University of Texas at Austin
Austin, TX, USA

**Quentin Rommel**
University of Texas at Austin
Austin, TX, USA

**Kevin S. Miller**
Brigham Young University
Provo, UT, USA

**Adam J. Thorpe**[*]
University of Texas at Austin
Austin, TX, USA

**Ufuk Topcu**
University of Texas at Austin
Austin, TX, USA

## ABSTRACT

Function encoders are a recent technique that learn neural network basis functions to form compact, adaptive representations of Hilbert spaces of functions. We show that function encoders provide a principled connection to feature learning and kernel methods by defining a kernel through an inner product of the learned feature map. This kernel-theoretic perspective explains their ability to scale independently of dataset size while adapting to the intrinsic structure of data, and it enables kernel-style analysis of neural models. Building on this foundation, we develop two training algorithms that learn compact bases: a progressive training approach that constructively grows bases, and a train-then-prune approach that offers a computationally efficient alternative after training. Both approaches use principles from PCA to reveal the intrinsic dimension of the learned space. In parallel, we derive finite-sample generalization bounds using Rademacher complexity and PAC-Bayes techniques, providing inference time guarantees. We validate our approach on a polynomial benchmark with a known intrinsic dimension, and on nonlinear dynamical systems including a Van der Pol oscillator and a two-body orbital model, demonstrating that the same accuracy can be achieved with substantially fewer basis functions. This work suggests a path toward neural predictors with kernel-level guarantees, enabling adaptable models that are both efficient and principled at scale.

## 1 INTRODUCTION

Learning methods face a persistent trade-off between computational efficiency and theoretical guarantees. Neural networks learn flexible representations and scale effectively to massive datasets, but their theoretical guarantees remain limited. Existing neural network bounds often rely on restrictive assumptions or yield vacuous estimates. Kernel methods provide precise statistical guarantees and well-developed theory, but scale poorly in practice. The source of this limitation lies in the dual formulation: kernel solutions are linear combinations of the training data, meaning inference cost is proportional to the number of training points $m$. Many applications, such as robotics and scientific modeling, demand both: scalable predictors that also come with theoretical guarantees.

We study function encoders, a recent technique in transfer and representation learning that bridges this gap by learning neural basis functions that act as explicit feature maps in the primal space (Ingebrand et al., 2025). Function encoders are trained using multiple datasets corresponding to multiple functions to learn a shared finite set of basis functions $\{\psi_j\}_{j=1}^n$ that define an explicit

---

[*]Corresponding author A. Thorpe, `adam.thorpe@austin.utexas.edu`.

feature map $\phi(x) = [\psi_1(x), \ldots, \psi_n(x)]^\top$. Any function in the span of these basis functions can be written as $\hat{f}(x) = \langle c, \phi(x) \rangle$ for some coefficient vector $c \in \mathbb{R}^n$, which is computed by solving a regularized least-squares problem. The key points are that the features are learned, not chosen, and inference cost depends only on the number of basis functions $n$, not the dataset size $m$. This makes function encoders computationally comparable to linear models in $n$ dimensions, while retaining the structure of kernel-based approaches.

Our results establish the theoretical role of function encoders. Function encoders can be understood from both the primal and dual perspectives. In the primal space, they provide an explicit feature map $\phi(x)$ that supports efficient linear prediction with cost $\mathcal{O}(n)$ per test point. In the dual space, the inner product $\langle \phi(x), \phi(x') \rangle$ corresponds to a kernel evaluation. This dual perspective makes it possible to analyze function encoders with the same theoretical tools used for kernels and design flexible neural training algorithms.

We present three key contributions: (1) We connect function encoders to feature learning and kernel methods by showing that function encoders define a kernel through the inner product of the learned feature map. This perspective unifies neural basis learning with kernel theory and explains why function encoders scale while retaining structure. (2) We develop two training algorithms based on principal component analysis (PCA) for learning compact bases: a progressive training approach that constructively grows bases and a train-then-prune approach that removes them after training. (3) We derive finite-sample generalization bounds using Rademacher complexity and PAC-Bayes analysis, giving inference-time guarantees for neural predictors. We validate our approach on an illustrative polynomial benchmark with a known intrinsic dimension, and demonstrate the capability of our approach on two nonlinear dynamical systems: a Van der Pol oscillator and a two-body model. Our results show that using our approach, we retain the same accuracy as an overparameterized model, but with significantly fewer basis functions.

## 2 RELATED WORK

**Function encoders:** Our work builds on the recent formulation of function encoders introduced in Ingebrand et al. (2025; 2024b;a). Function encoders have applications in robotics (Ward et al., 2025a;b), dynamics modeling (Ingebrand et al., 2024a), and transfer learning (Ingebrand et al., 2024b; 2025), but existing analysis is limited. Prior work established asymptotic results (Ingebrand et al., 2025, Theorem 1) showing that as $n \to \infty$ the span of these basis functions can represent the entire Hilbert space. However, theoretical gaps remain in the existing work: (i) there is no formal connection between function encoders and Hilbert space techniques, (ii) there are no principled methods to choose the number of basis functions $n$, and (iii) existing work lacks finite sample guarantees that quantify the quality of the approximation once the basis functions are learned. We extend the existing foundation to fill this gap.

**Kernel methods and kernel approximation:** Kernel methods such as Gaussian processes, kernel ridge regression, and support vector machines provide strong guarantees through RKHS theory and often support closed-form training (Schölkopf & Smola, 2002; Christmann & Steinwart, 2008). Despite this, two persistent challenges limit their use in practice: scalability and kernel choice. Approximate kernel methods aim to address these limitations. Nyström sampling (Drineas & Mahoney, 2005; Rudi et al., 2015) reduces the computational burden of large kernel matrices by subsampling columns to construct a low-rank approximation. However, the quality of this approximation depends heavily on the initial kernel and the choice of sampling strategy. Random Fourier features (Rahimi & Recht, 2007; Rudi & Rosasco, 2017) and Fastfood (Le et al., 2013) offer randomized approximations for shift-invariant kernels, constructing explicit feature maps from a fixed distribution, with provable generalization guarantees when the number of features is scaled appropriately. While these methods improve scalability by operating in the primal, they still require the kernel to be specified in advance. The feature map remains static, and no adaptation occurs during training, limiting their ability to generalize across diverse tasks or distributions. Function encoders differ by learning a data-dependent feature map in the primal that can be viewed as an adaptive, kernel-like predictor.

**Kernel learning:** Deep kernel learning adapts kernels by parameterizing them with a neural network $f_\theta$, giving $k_\theta(x, x') = k(f_\theta(x), f_\theta(x'))$, where $k$ is a base kernel such as RBF (Wilson et al., 2016). This improves expressiveness but retains the computational bottlenecks of kernel methods: inference still requires evaluating against all $m$ training points in the dual. Recent work has

established generalization guarantees for deep kernels through capacity analysis and deep kernel regression (Zhang & Zhang, 2023; Ji & Fu, 2024) that complement our bounds for function encoders. More recent work on implicit kernel learning (Li et al., 2019; Falk et al., 2022) learns the underlying spectral distribution with a neural generator, so the kernel is only defined implicitly through samples. As a result, these methods do not yield an explicit kernel function or a finite-dimensional basis of learned features. Kernel Flows and related approaches (Owhadi & Yoo, 2019; Yoo & Owhadi, 2021) similarly start from a fixed base kernel and learn input transformations or inner network layers so that the induced kernel better matches the data, but the associated RKHS is still determined a priori by the base kernel, rather than being learned from data. By contrast, function encoders are trained across multiple underlying functions and learn an explicit finite-dimensional feature map $\phi(x)$ that spans a shared function space. This feature map supports inference with cost depending only on the number of basis functions $n$, eliminating dataset-size dependence at inference while still enabling kernel-style analysis.

**Dictionary learning:** Dictionary learning methods K-SVD (Aharon et al., 2006), matching-pursuit (Mallat & Zhang, 1993), SINDy (Brunton et al., 2016), and Koopman approaches (Mezić, 2005; Williams et al., 2015) also seek compact representations by combining basis elements with coefficients. Neural variants (Lee et al., 2022; Lusch et al., 2018; Takeishi et al., 2017) learn dictionaries or observables jointly with the model and resemble the neural basis learning of function encoders. These methods, however, have fundamentally different objectives and problem settings: sparsity for model discovery (SINDy, dictionaries, K-SVD, matching pursuit) or linearization of dynamics (Koopman). Their guarantees typically concern sparsity recovery or consistency. Function encoders instead learn basis functions to span a subspace across tasks, followed by ridge regression per task, and admit RKHS-style prediction bounds.

## 3 FUNCTION ENCODERS AS LEARNED FEATURE REPRESENTATIONS

Function encoders can be viewed as feature maps learned through neural basis functions. The function encoder optimization exactly matches the primal feature learning problem in a Hilbert space $\mathcal{H}$. For the purpose of illustration, we first consider the scalar case, and show that function encoders naturally extend to the vector-valued case in Appendix B. Let $\mathcal{H}$ be a Hilbert space. A function encoder learns a set of basis functions $\{\psi_j\}_{j=1}^n$, which together define a feature map $\phi : \mathcal{X} \to \mathbb{R}^n$,

$$\phi(x) = [\psi_1(x), \ldots, \psi_n(x)]^\top, \tag{1}$$

so that any function $f$ in the span of these basis functions can be written as $f(x) = \langle c, \phi(x) \rangle$ for some coefficient vector $c \in \mathbb{R}^n$. Given training data $(x_1, y_1), \ldots, (x_m, y_m)$, the coefficients $c$ of the function approximation $\hat{f}$ are computed at inference time by solving a regularized least-squares problem,

$$\min_{c \in \mathbb{R}^n} \frac{1}{m} \sum_{i=1}^m (y_i - \langle c, \phi(x_i) \rangle)^2 + \lambda \|c\|^2, \tag{2}$$

which has a closed-form solution,

$$\left( \frac{1}{m} \sum_{i=1}^m \phi(x_i) \phi(x_i)^\top + \lambda I \right) c = \frac{1}{m} \sum_{i=1}^m y_i \phi(x_i). \tag{3}$$

A function encoder learns in two stages. In the offline phase, the function encoder is trained on a collection of $N$ datasets $\{D_1, \ldots, D_N\}$, where each $D_j = \{(x_i, f_j(x_i))\}_{i=1}^m$ comes from a different underlying ground-truth function $f_j \in \mathcal{H}$. For each $f_j$, the function encoder produces an approximation $\hat{f}_j(x) = \langle c_j, \phi(x) \rangle \in F_n$, where $F_n = \text{span}\{\psi_1, \ldots, \psi_n\} \subset \mathcal{H}$. The function encoder minimizes the loss,

$$\frac{1}{Nm} \sum_{j=1}^N \sum_{i=1}^m \|f_j(x_i) - \hat{f}_j(x_i)\|^2 + \lambda \|\hat{f}_j\|^2, \tag{4}$$

by alternating between solving for each $c_j$ via regularized least squares with the current feature map and updating the basis parameters by gradient descent. After training, the basis functions $\{\psi_j\}_{j=1}^n$ are fixed. Then at inference time, we can compute and update the coefficients $c$ using a small amount

of online data via least squares (2). Standard neural networks learn a single predictor. In contrast, function encoders learn a reusable basis and solve task-specific linear problems within that basis. See Ingebrand et al. (2025) for more details.

Note that a particular set of basis functions learned is not unique and can vary across training runs, since multiple distinct, valid bases can span the same function space. Moreover, within a set, the basis functions need not be orthonormal. Since we compute the coefficients via least squares, we only require the basis functions to be linearly independent. To enforce orthonormality, it is possible to use the Gram-Schmidt process during training, but this significantly increases training time. In practice, regularization or soft penalties are preferable to keep the bases well-conditioned without incurring large computational overhead.

## 3.1 Function Encoders Are Learnable Kernels

Function encoders define a kernel $k : \mathcal{X} \times \mathcal{X} \to \mathbb{R}$ through the inner product, which is automatically a valid (symmetric, positive semi-definite) kernel.

**Proposition 1.** *Let $\phi(x) = [\psi_1(x), \ldots, \psi_n(x)]^\top$. The kernel $k$, defined by*

$$k(x, x') = \langle \phi(x), \phi(x') \rangle = \sum_{j=1}^{n} \psi_j(x) \psi_j(x'). \tag{5}$$

*is a valid reproducing kernel.*

*Proof.* For any $\alpha \in \mathbb{R}^m$ and $\{x_i\}_{i=1}^m$, $\|\sum_i \alpha_i \phi(x_i)\|^2 \geq 0$, and $k(x, x') = k(x', x)$ by symmetry of the inner product. Thus, $k$ is symmetric positive semi-definite, and therefore a valid kernel. □

We note that the kernel is induced by a learned feature map. We learn the finite-dimensional RKHS itself (a compact, shared subspace across tasks) via neural basis learning. The induced kernel $k(x, x') = \langle \phi(x), \phi(x') \rangle$ is a byproduct, not a design choice. This contrasts with Nyström and random Fourier features, which approximate a pre-specified kernel, and with Deep Kernel Learning, which still operates in a fixed RKHS and incurs $\mathcal{O}(m)$ inference.

In the dual space, learning is expressed in terms of kernel evaluations between data points. By the representer theorem (Schölkopf et al., 2001), the optimal solution to the primal problem (2) lies in the span of features over the training data. In particular, with $m$ training points, the solution can be expressed as $w^* = \sum_{j=1}^m \alpha_j \phi(x_j)$. Substituting this into the primal objective function in (2) yields

$$\min_{\alpha \in \mathbb{R}^m} \frac{1}{m} \sum_{i=1}^{m} \left( y_i - \sum_{j=1}^{m} \alpha_j k(x_i, x_j) \right)^2 + \lambda \|w^*\|^2. \tag{6}$$

The solution can be computed as the solution to the linear system $(K + \lambda m I)\alpha = Y$, where $K_{ij} = k(x_i, x_j)$ and $Y = [y_1, \ldots, y_m]^\top$. Solving the primal problem (2) is more efficient when $n \ll m$, while solving the dual (6) is preferable when $m \ll n$. This shows that function encoders not only provide explicit feature maps in the primal, but also instantiate valid kernels in the dual.

## 4 Learning Compact Feature Representations

A central question for function encoders is how many basis functions are needed for a given dataset. Existing theory ensures completeness as $n \to \infty$, but there is currently no principled rule for selecting a compact set of basis functions. Without such a rule, models risk either underfitting (too few basis functions) or overfitting, which leads to redundancy and inefficiency (too many). We develop two PCA-guided training strategies to address this question and identify compact bases: a sequential, progressive training approach and a parallel train-then-prune approach.

### 4.1 Progressive Training of Basis Functions

Progressive training builds the basis set sequentially, ensuring each new function captures variance not explained by the previous ones. This method explicitly leverages PCA in the space of coefficients and offers interpretability, but at the cost of sequential training.

Progressive training learns the basis functions one at a time, starting with a single basis function. After training, the basis function is frozen. We then create a new basis function, add it to the basis set, and repeat training on the new basis function. In particular, at step $b = 1$, we train a single basis $\psi_1$ using the function encoder objective in (4). After training, we freeze $\psi_1$. At step $b = 2$, we add a second basis $\psi_2$ to form $\phi_2(x) = [\psi_1(x), \psi_2(x)]^\top$. We compute coefficients $c^j$ for each dataset $D_j$ via (3). During optimization, we update only the parameters of $\psi_2$ so that it effectively fits the residual $f_j(x) - \langle c^j, \psi_1(x) \rangle$. At each step $b$, the new basis $\psi_b$ is trained while $\{\psi_1, \ldots, \psi_{b-1}\}$ remain fixed. While freezing the previous basis functions is not strictly necessary, it means each new basis captures variance not explained by the frozen bases and prevents collapse or redistribution of variance among earlier bases. This creates a natural ordering of basis functions analogous to PCA.

We then compute the coefficients $\{c^1, \ldots, c^N\}$ across all training datasets $\{D_1, \ldots, D_N\}$ and form the mean-centered covariance matrix,

$$\Sigma_b = \frac{1}{N-1} \sum_{i=1}^{N} (c^i - \bar{c})(c^i - \bar{c})^\top. \tag{7}$$

Let $\lambda_1 \geq \ldots \geq \lambda_b$ be the eigenvalues. The cumulative explained variance, $\text{CEV}_r = \sum_{i=1}^{r} \text{EVR}_i$, where $\text{EVR}_k = \lambda_k / \sum_{i=1}^{b} \lambda_i$ is the explained variance ratio, gives a principled proxy for the effective dimension of the space. Training stops once $\text{CEV}_r \geq \tau$ for a user-specified threshold $\tau$ (e.g. 99%). This rule selects the effective rank of the coefficient covariance, which serves as a proxy for the intrinsic dimension of the data. PCA is used in exactly this same fashion, where the eigenvalue spectrum often shows a sharp elbow once the bases span the intrinsic dimension.

The main limitation is that training is inherently sequential in $b$, which prevents parallelization on GPUs. The primary benefit is an ordered, interpretable basis with a clear stopping rule that signals when added capacity no longer improves representation quality. As in PCA, the stopping threshold remains primarily heuristic, however. We summarize the training loop as Algorithm 1.

---

**Algorithm 1** Progressive Training

---

**Require:** Datasets $\{D_j\}_{j=1}^{N}$, variance threshold $\tau$
 1: Initialize $\mathcal{B} \leftarrow \varnothing$, $\text{CEV} \leftarrow 0$, $b \leftarrow 0$
 2: **while** $\text{CEV} < \tau$ **do**
 3:      $b \leftarrow b + 1$; add new basis $\psi_b$ to $\mathcal{B}$; freeze $\{\psi_1, \ldots, \psi_{b-1}\}$
 4:      Train $\psi_b$
 5:      **for** each dataset $D_j$ **do**
 6:          Compute coefficients $c^j$ via (3)
 7:      **end for**
 8:      Collect $\{c^j\}_{j=1}^{N}$, compute $\Sigma_b$
 9:      Update CEV from eigenvalues of $\Sigma_b$
10: **end while**
11: **return** $\mathcal{B} = \{\psi_1, \ldots, \psi_b\}$

---

## 4.2 TRAIN-THEN-PRUNE

Train-then-prune takes advantage of parallel computation. This approach is more computationally efficient, but requires careful pruning and retraining. Instead of building bases sequentially, we overparameterize with $B$ bases $\{\psi_1, \ldots, \psi_B\}$ and train them jointly using (4). We then compute the coefficients for each function in the datasets and compute the covariance matrix $\Sigma_B$ as in (7). From the eigenvalues of $\Sigma_B$, we compute the effective rank,

$$r = \min\left\{ n : \frac{\sum_{i=1}^{n} \lambda_i}{\sum_{j=1}^{B} \lambda_j} \geq \tau \right\}, \tag{8}$$

which is the minimum number of basis functions required to capture at least $\tau$ of the variance.

Because the bases are trained jointly, they are not naturally ordered. To prune the basis functions, we need to select the $r$ most informative basis functions. We score each basis $\psi_p$ by

$$s_p = \sum_{i=1}^{r} \lambda_i U_{pi}^2, \tag{9}$$

where $U$ contains eigenvectors of $\Sigma_B$. Alternative scoring rules, such as cosine similarity between bases and eigenvectors, ignore eigenvalue magnitudes and yield higher reconstruction error.

We then keep the top-$r$ basis functions and prune the rest. In a multi-headed MLP, this corresponds to removing parameters from the final layer to form a reduced-size network. Unlike the progressive training algorithm, the basis functions selected by the train-then-prune algorithm are not guaranteed to capture the desired variance. We then perform a short fine-tuning step to retrain the basis functions to capture the residual variance. We summarize the procedure in Algorithm 2.

---

**Algorithm 2** Train-Then-Prune

---

**Require:** Datasets $\{D_j\}_{j=1}^N$, initial $B \gg r$, variance threshold $\tau$
 1: Initialize function encoder with $\{\psi_1, \ldots, \psi_B\}$; train jointly on all tasks
 2: **for** each dataset $D_j$ **do**
 3:     Compute coefficients $c^j$ via (3) with $\phi_B$
 4: **end for**
 5: Form covariance $\Sigma_B$, eigendecompose to $(U, \lambda_i)$
 6: Compute $r = \min\{n : \sum_{i=1}^n \lambda_i / \sum_{j=1}^B \lambda_j \geq \tau\}$
 7: Score each basis $s_p = \sum_{i=1}^r \lambda_i U_{pi}^2$
 8: Keep top-$r$ bases, prune others
 9: Fine-tune the reduced model
10: **return** Top-$r$ bases

---

The basis-selection procedures developed in this section determine a compact feature dimension $n$ using the CEV of the coefficient covariance. This selected dimension $n$ shows up explicitly in the complexity terms that appear in our generalization bounds in the next section, which scale on the order of $\sqrt{n/(m\lambda)}$. Although our analysis does not explicitly model the CEV, the CEV threshold thus acts as a data-driven estimate of the intrinsic dimension that determines the size of this generalization term. Making this connection fully rigorous would require relating the empirical spectrum of the coefficient covariance to approximation error, which we leave for future work.

## 5    DETERMINING COMPLEXITY AND GENERALIZATION BOUNDS

The bounds derived in this section depend explicitly on the neural network feature map and, in particular, on the number of learned basis functions $n$ chosen by the CEV-based procedures in Section 4. Once the basis functions are fixed, a function encoder reduces to ridge regression in a $n$-dimensional feature space. The central question then becomes: how does generalization depend on $n$, the sample size $m$, and regularization $\lambda$? We address this by analyzing the complexity of the induced hypothesis class using two complementary analyses: Rademacher complexity and PAC-Bayes.

### 5.1    RADEMACHER COMPLEXITY BOUNDS

The Rademacher complexity measures how "rich" a function class is (Bartlett & Mendelson, 2003). A high Rademacher complexity indicates that a function class is able to closely model more complex functions. Intuitively, the Rademacher complexity helps quantify the balance between model expressiveness and the ability to generalize to unseen data.

Let $\hat{c}_\lambda$ denote the solution to (2) with regularization parameter $\lambda > 0$,

$$\hat{c}_\lambda := \arg\min_{c \in \mathbb{R}^n} \frac{1}{m} \sum_{i=1}^m (\langle \phi(x_i), c \rangle - y_i)^2 + \lambda \|c\|_2^2, \tag{10}$$

with corresponding loss function of interest $\ell(f_c(x), y) = (f_c(x) - y)^2 = (\langle c, \phi(x) \rangle - y)^2$. Define the population risk $L(f_c) := \mathbb{E}[\ell(f_c(x), y)]$. Then, given an empirical sample of i.i.d. data $\mathcal{S} := \{(x_i, y_i)\}_{i=1}^m \subset \mathcal{X} \times \mathcal{Y}$, define the empirical risk $\hat{L}_m(f_c) = \frac{1}{m} \sum_{i=1}^m \ell(f_c(x_i), y_i)$.

Under some mild assumptions on the boundedness of the basis functions $\{\psi_1, \ldots, \psi_n\}$ and outputs $y \in \mathcal{Y}$, we have that the learning scheme of (2) satisfies the following result:

**Theorem 1.** *Let $\psi_1, \ldots, \psi_n \subset \mathcal{H}$ be a* fixed *set of basis functions, where each $\psi_j : \mathcal{X} \to \mathcal{Y}$ is a fixed, bounded neural network satisfying $\sup_{x \in \mathcal{X}} |\psi_j(x)| \leq R$. Assume that the output space for the regularized least-squares problem of* (2) *is uniformly bounded as $\sup_{y \in \mathcal{Y}} \|y\|_2 \leq Y$. Given regularization parameter $\lambda > 0$, then for any $\delta > 0$ we have that with probability greater than or equal to $1 - \delta$ the least-squares solution $f_{\hat{c}_\lambda}$ from* (10) *satisfies*

$$L(f_{\hat{c}_\lambda}) \leq \hat{L}_m(f_{\hat{c}_\lambda}) + 2Y^2 R \sqrt{\frac{n}{m\lambda}} \left( R\sqrt{\frac{n}{\lambda}} + 1 \right) \left( 2 + \sqrt{\frac{\log(1/\delta)}{2}} \right) \tag{11}$$

$$\lesssim \hat{L}_m(f_{\hat{c}_\lambda}) + \tilde{\mathcal{O}}\left( Y^2 R^2 \frac{n}{\lambda \sqrt{m}} \right). \tag{12}$$

The proof follows from the Rademacher complexity of regularized linear predictors (Kakade et al., 2008), and is presented in Appendix C.1. The key takeaway is the scaling: complexity grows with the number of bases $n$, but decreases with data size $m$ and regularization $\lambda$. Thus, more bases increase expressivity but also the risk of overfitting unless compensated for by sufficient data or stronger regularization.

## 5.2   PAC BAYES

PAC-Bayes analysis provides probabilistic guarantees that hold for randomized predictors, but applying it to function encoders is not straightforward. Existing results typically assume fixed features, scalar outputs, and bounded loss functions, whereas function encoders involve learned, multivariate bases and regularized least-squares coefficients.

To address this, we overcome two primary challenges: we work in the fixed basis setting and use truncated Gaussian distributions for the prior and posterior to handle the unbounded loss function in the regularized least squares problem. This extension is non-trivial, and handles multivariate outputs, accommodates learned feature maps, and yields non-vacuous guarantees. More broadly, the techniques for controlling the KL divergence between truncated Gaussians extend beyond our setting, offering a general-purpose tool for PAC-Bayes analysis for unbounded loss function settings.

For the fixed basis setting, we obtain:

**Theorem 2.** *Let $\psi_1, \ldots, \psi_n \subset \mathcal{H}$ be a* fixed *set of basis functions, where each $\psi_j : \mathcal{X} \to \mathcal{Y}$ is a fixed, bounded neural network satisfying $\sup_{x \in \mathcal{X}} |\psi_j(x)| \leq R$. Assume furthermore that $\mathcal{Y}$ is uniformly bounded as $\sup_{y \in \mathcal{Y}} \|y\|_2 \leq Y$ and that the mapping $\Phi(c) = \sum_{i=1}^n c_i \psi_i$ is injective. Given regularization parameter $\lambda > 0$, then for any $\delta > 0$ we have that with probability greater than or equal to $1 - \delta$ the least-squares solution $f_{\hat{c}_\lambda}$ from* (10) *satisfies*

$$L(f_{\hat{c}_\lambda}) \lesssim \hat{L}_m(f_{\hat{c}_\lambda}) + \tilde{\mathcal{O}}\left( Y^2 R^2 \frac{n^{3/2}}{\lambda \sqrt{m}} \right). \tag{13}$$

The proof is presented in Appendix C.2. This result highlights the stability of the predictor under posterior perturbations, aligning with Bayesian interpretations of kernel ridge regression.

It is worth noting that the two bounds cover complementary aspects. Rademacher bound is distribution free, while PCA-Bayes yields a tighter bound if and only if the prior is data dependent. The PAC-Bayes bound is looser in our current presentation only because we use a truncated Gaussian chosen only to handle unbounded losses. Theorem 2 establishes this framework for future work.

The basis-selection procedures developed in Section 4 constructs the feature map of rank $n_\tau$. After pruning, the basis $\{\psi_1, \ldots, \psi_{n_\tau}\}$ is fixed and defines an $n_\tau$-dimensional hypothesis class $F_{n_\tau}$. The finite-sample bounds of Theorem 1 and Theorem 2 defined in Section 5 apply to any fixed feature map, so substituting $n = n_\tau$ yields estimation terms $\tilde{\mathcal{O}}\left( \frac{n_\tau}{\sqrt{m}\lambda} \right)$ and $\tilde{\mathcal{O}}\left( \frac{n_\tau^{3/2}}{\sqrt{m}\lambda} \right)$. Consequently, the PCA-guided procedures explicitly select the complexity parameter that governs the generalization behavior in Section 5.

Taken together, Theorems 1 and 2 suggest that, from a theoretical standpoint, we should use the smallest number of basis functions that still captures the structure of the data. The PCA-guided procedures in Section 4 provide a practical mechanism for identifying this dimension.

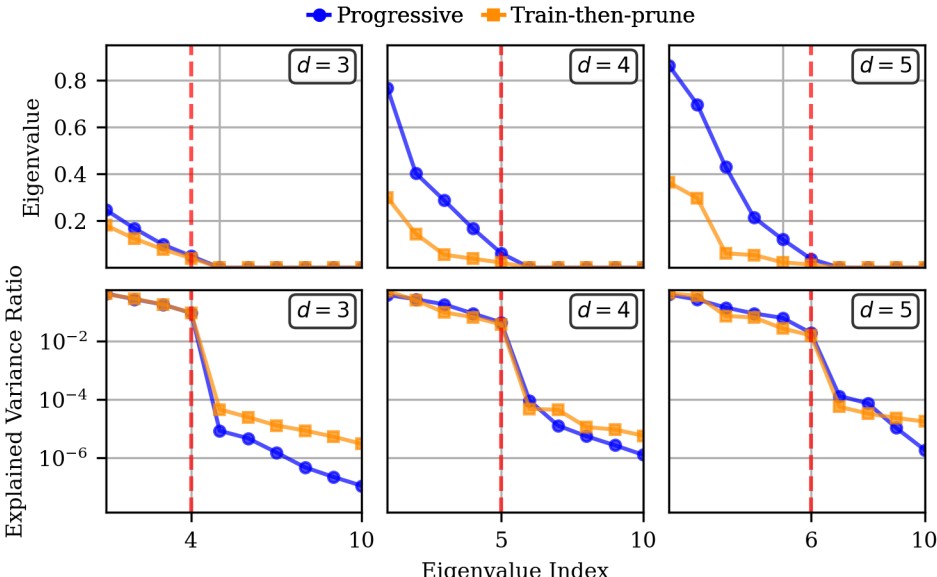

Figure 1: Function encoders recover the intrinsic polynomial dimension using our proposed algorithms. Scree plots of the coefficient covariance (top row) show rapid eigenvalue decay with a clear elbow at the same cutoff, confirming that both the loss curves and variance analysis identify the correct number of bases. The explained variance ratio of the eigenvalues (bottom row) shows a sharp drop when the intrinsic dimension $(d + 1)$ is reached.

## 6 EXPERIMENTAL RESULTS

We evaluate our approach on an illustrative polynomial benchmark with a known, finite intrinsic dimension, and on two dynamical systems examples to showcase our approach: a Van der Pol oscillator and a planar two-body orbital model. The polynomial benchmark serves as a controlled setting where the intrinsic dimension is known, while the dynamical system examples extend our approach to practical nonlinear dynamics modeling problems of practical relevance in robotics and orbital mechanics. We train all benchmarks using $N = 1000$ functions.

### 6.1 AN ILLUSTRATIVE EXAMPLE ON POLYNOMIAL SPACES

We first consider an illustrative benchmark to validate our two proposed algorithms on polynomial spaces of varying degrees $d \in \{3, 4, 5\}$, where the intrinsic dimension is $d + 1$. Scree plots of the coefficient covariance matrix in Fig. 1 show rapid eigenvalue decay with clear elbows at the expected dimensionality. Both approaches recover the correct number of basis functions: four basis functions for degree-3 polynomials (Fig. 5 and 6 in Appendix D.1), five for degree-4, and six for degree-5. The train-then-prune approach selects the same cutoff points, and fine-tuned pruned models achieve reconstruction accuracy that is identical to the original overparameterized networks. The progressive algorithm training curves initially show sharp reductions in mean squared error, followed by a plateau once the intrinsic dimension is reached (Fig. 4 in Appendix D.1). We evaluated both a multi-headed MLP architecture that is more computationally efficient since it shares hidden parameters across the basis functions, as well as a basis specified by independent MLPs.

#### 6.1.1 COMPARISON AND CONNECTIONS WITH DEEP KERNELS

For comparison, we train an RBF deep kernel (Wilson et al., 2016) on the same space of degree-3 polynomials. Deep kernels are designed to adapt a kernel for a single supervised task, whereas our setting requires learning a function space across many training functions. We adapt the deep kernel training to our setting for a direct comparison. Details are provided in Appendix D.1.1. With $m = 20$ evaluation points, the deep kernel takes about ten times longer than a comparable function encoder

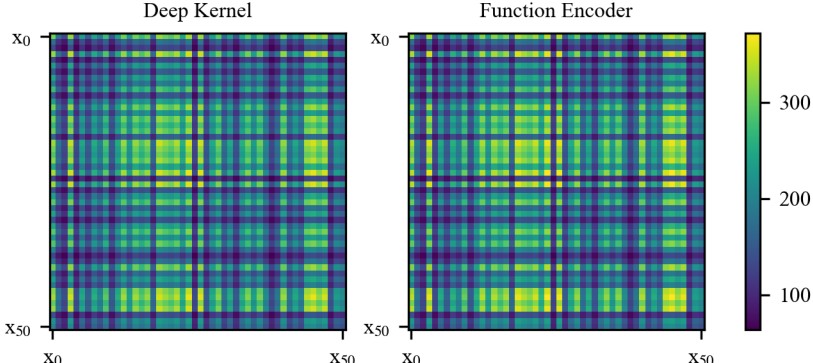

Figure 2: Gram matrices from a deep kernel (left) and a function encoder (right) on degree-3 polynomials. Both yield nearly identical geometry, but function encoders obtain it with a fixed-size primal formulation, while deep kernels require cubic-cost Gram matrix inversions in $m$.

to reach the same MSE $\approx 10^{-6}$. The slowdown arises from the $m \times m$ Gram matrix inversion in the dual formulation, which introduces a per-function-per-batch cost of $\mathcal{O}(m^3)$ that accumulates across training functions. In contrast, the function encoder solves a fixed-size least-squares problem in the primal, so its training time only grows with $n$ and is independent of $m$. At inference, runtimes are similar as both models use comparable network architectures. For $m = 20$ and $n = 4$, the one-time coefficient estimation cost is negligible. The main difference comes from the prediction step, where deep kernels rely on kernel evaluations and function encoders on basis evaluations. Thus, despite the comparable inference time (basis vs. kernel evaluations), the function encoder avoids the $\mathcal{O}(m^3)$ training overhead, resulting in the $\approx 10\times$ speedup observed for $m = 20$ and $n = 4$.

We compare function encoders with a linear deep kernel to examine the induced geometry. Training with the linear deep kernel remains slower than with function encoders, though faster than with an RBF kernel (roughly twice as fast in our setup), highlighting that the nonlinear kernel in deep kernel training directly impacts scalability. Increasing the example set to $m = 50$ does not change training time. Figure 2 shows that the Gram matrices from both approaches have nearly identical structure. This is consistent with theory: a function encoder can be viewed as a deep linear kernel trained in the primal, recovering the same geometry up to rescalings of the basis. The key difference is efficiency since function encoders achieve this geometry more effectively when $m \gg n$.

## 6.2 MODELING DYNAMICAL SYSTEMS WITH NEURAL ODE BASIS FUNCTIONS

We next evaluate our approach on two dynamical systems examples, where compact feature representations are critical for real-time robotics, control, and autonomous systems. We focus on the Van der Pol oscillator to compare with prior work in Ingebrand et al. (2025; 2024a) and the planar two-body system to demonstrate our approach on a challenging, real-world satellite orbit prediction problem. We implement basis functions as neural ODEs (Ingebrand et al., 2024a), which are useful for capturing long-term dynamical behaviors.

For both tasks, we generate trajectories by sampling initial conditions from a bounded region of the state space. We then use an RK4 integration scheme to compute the trajectories to form datasets $D_i$ with data of the form $(x_t, \Delta t, x_{t+1} - x_t)$ (c.f. Ingebrand et al., 2024a).

### 6.2.1 VAN DER POL

The Van der Pol oscillator is a nonlinear system with nontrivial limit-cycle behavior. Prior work in Ingebrand et al. (2024b) used 100 neural ODE basis functions to model the space of dynamics. However, our results show that only 2 basis functions are sufficient to capture the space. Both the progressive training algorithm (Algorithm 1) and train-then-prune (Algorithm 2) identify the same cutoff point. We see a sharp decline in the explained variance ratios after two basis functions, and the training loss plateaus once two basis functions are trained. Despite reducing the number of bases by an order of magnitude, the compact representation achieves the same predictive accuracy as the

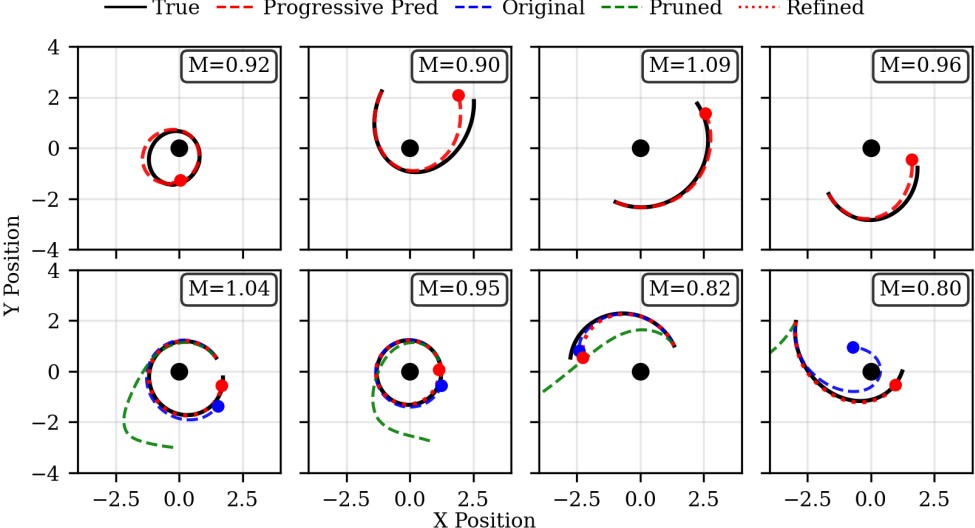

Figure 3: The first row shows the progressive training, where the predicted trajectories follow ground truth orbits using only five to six bases. The second row shows the comparison of the overparameterized model (blue), the pruned model (green), and the refined version (red) against the ground truth (black). The pruned and refined model meets or surpasses the original model after retraining.

original overparameterized encoder. This indicates that most of the additional bases in prior work are redundant. While they do not degrade accuracy, they add unnecessary computational overhead.

### 6.2.2 TWO-BODY PROBLEM

We consider the normalized planar two-body problem with a gravitational parameter. Initial conditions are sampled to yield bound elliptical orbits. The planar two-body setting serves as a more demanding benchmark because, unlike the Van der Pol oscillator, which has a single low-dimensional attractor, it is a conservative system with a continuous family of elliptical orbits determined by energy and angular momentum. This diversity leads to longer-term correlations and makes the function space effectively higher-rank. Consequently, the effective rank is not sharply defined: the eigenvalue spectra of the coefficient covariance decay more gradually.

In the planar two-body setting (Fig. 3), the space of possible orbits is five-dimensional: one dimension for the central mass $\mu$, and four for the orbital parameters $(a, e, \omega, \nu)$ describing size, shape, in-plane orientation, and phase. Consistent with this structure, the explained variance analysis shows that five to six bases capture more than 99% of the variance. This is consistent with the gradual eigenvalue decay in the scree plot. Unlike the Van der Pol oscillator, where variance concentrates in a few dominant modes, the two-body system requires additional bases to represent its richer dynamics. Streamplots of the learned bases reveal interpretable structures aligned with orbital dynamics, highlighting that the bases are compact and physically meaningful.

The key takeaway is that function encoders adapt to the complexity of the system. For Van der Pol, they uncover a simple two-dimensional structure. For the two-body problem, they scale up to model more intricate behavior while still yielding a compact and informative representation. This is especially valuable for real-world applications such as embedded controllers, onboard satellite orbit determination, or autonomous navigation, where compact models that also offer guarantees of correctness are required under strict computational limits.

## 7 CONCLUSION & FUTURE WORK

We develop a principled connection between function encoders, neural models that learn compact feature maps, and RKHS theory. Our contributions include PCA-guided algorithms for selecting

compact bases and finite-sample generalization bounds that extend kernel-style analysis to neural predictors. Function encoders combine the efficiency of parametric models with the rigor of kernel methods, enabling scalable yet principled learning. Several open directions remain: developing non-heuristic criteria for basis selection, deepening theoretical links with kernel methods, and extending the framework to applications such as statistical and scientific modeling. These opportunities point toward a broader role for function encoders as efficient neural models with kernel-level guarantees.

## 8 Reproducibility Statement

All code, data generation scripts, and hyperparameter settings are available at `https://github.com/suann124/function_encoder_kernels`. Additional implementation details and results are provided in the appendix. Proofs of the theoretical results, including Rademacher complexity and PAC-Bayes bounds, are provided in the appendix with explicit assumptions stated. For experiments, we describe the polynomial benchmark, Van der Pol oscillator, and two-body problem in detail in the appendix. Additional diagnostic plots and training details are also included.

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

## A  THE USE OF LARGE LANGUAGE MODELS (LLMS)

We used a large language model as a general-purpose writing and editing assistant. Its role was limited to suggesting alternative phrasings, improving clarity and flow, and helping with the structural organization of the manuscript. All technical ideas, theoretical results, proofs, algorithms, and experiments were developed entirely by the authors.

## B   THE VECTOR-VALUED CASE

Many tasks require vector-valued outputs (e.g., multi-output regression and dynamical systems). We extend function encoders to the vector-valued setting. Let $\mathcal{Y} = \mathbb{R}^d$ be the output space and consider vector-valued basis functions $\psi_j : \mathcal{X} \to \mathcal{Y}$ for $j = 1, \ldots, n$. The feature map $\phi : \mathcal{X} \to \mathbb{R}^{d \times n}$ is given by,

$$\phi(x) = [\psi_1(x), \ldots, \psi_n(x)]. \tag{14}$$

A predictor is of the form

$$\hat{f}(x) = \phi(x)c = \sum_{j=1}^{n} c_j \psi_j(x), \tag{15}$$

where $c \in \mathbb{R}^n$.

Given training data $(x_1, y_1), \ldots, (x_m, y_m)$ with $y_i \in \mathbb{R}^d$, the regularized least-squares problem is

$$\min c \in \mathbb{R}^n \frac{1}{m} \sum_{i=1}^{m} \|y_i - \phi(x_i)c\|_2^2 + \lambda \|c\|_2^2. \tag{16}$$

The normal equations are

$$\left( \frac{1}{m} \sum_{i=1}^{m} \phi(x_i)^\top \phi(x_i) + \lambda I_n \right) c = \frac{1}{m} \sum_{i=1}^{m} \phi(x_i)^\top y_i \in \mathbb{R}^n. \tag{17}$$

This setting induces an operator-valued kernel (Micchelli & Pontil, 2005),

$$\kappa(x, x') = \sum_{j=1}^{n} \psi_j(x) \psi_j(x')^\top. \tag{18}$$

In matrix form, the predictor can be expressed as,

$$\hat{f}(x) = \sum_{i=1}^{m} \kappa(x, x_i) \alpha_i, \tag{19}$$

where the coefficients $\alpha$ are found as the solution to the linear system, $(K + \lambda m I_m)\alpha = Y$, where $Y = [y_1, \ldots, y_m] \in \mathbb{R}^{d \times m}$ and $K$ is the Gram matrix with blocks $K_{ij} = \kappa(x_i, x_j)$.

## C   GENERALIZATION BOUNDS

### C.1   PROOF OF THEOREM 1

Here we provide a proof of Theorem 1, restated below.

**Theorem 1.** *Let $\psi_1, \ldots, \psi_n \subset \mathcal{H}$ be a* fixed *set of basis functions, where each $\psi_j : \mathcal{X} \to \mathcal{Y}$ is a fixed, bounded neural network satisfying $\sup_{x \in \mathcal{X}} |\psi_j(x)| \leq R$. Assume that the output space for the regularized least-squares problem of (2) is uniformly bounded as $\sup_{y \in \mathcal{Y}} \|y\|_2 \leq Y$. Given regularization parameter $\lambda > 0$, then for any $\delta > 0$ we have that with probability greater than or equal to $1 - \delta$ the least-squares solution $f_{\hat{c}_\lambda}$ from (10) satisfies*

$$L(f_{\hat{c}_\lambda}) \leq \hat{L}_m(f_{\hat{c}_\lambda}) + 2Y^2 R \sqrt{\frac{n}{m\lambda}} \left( R\sqrt{\frac{n}{\lambda}} + 1 \right) \left( 2 + \sqrt{\frac{\log(1/\delta)}{2}} \right) \tag{11}$$

$$\lesssim \hat{L}_m(f_{\hat{c}_\lambda}) + \tilde{\mathcal{O}} \left( Y^2 R^2 \frac{n}{\lambda \sqrt{m}} \right). \tag{12}$$

This result follows straightforwardly from well-known results from (Kakade et al., 2008) that identify the corresponding Rademacher complexity when using regularization.

*Proof.* We leverage the result of Theorem 3 and Corollary 5 in (Kakade et al., 2008) to obtain generalization bounds for our regularized least-squares setting (2). To apply these results, we begin by establishing a few simple bounds. First, note that the least-squares solution from (10) can be bounded as

$$\lambda \|\hat{c}_\lambda\|_2^2 \leq \hat{L}_m(f_{\hat{c}_\lambda}) + \lambda \|\hat{c}_\lambda\|_2^2 \leq \hat{L}_m(f_0) + 0 = \frac{1}{m} \sum_{i=1}^m y_i^2 \leq Y^2 \tag{20}$$

$$\implies \|\hat{c}_\lambda\|_2 \leq \frac{Y}{\sqrt{\lambda}}, \tag{21}$$

where we have used the boundedness of the outputs $y_i \in \mathcal{Y}$. Define the relevant set of coefficients as

$$\mathcal{C}_\lambda := \{c \in \mathbb{R}^n : \|c\|_2 \leq \frac{Y}{\sqrt{\lambda}}. \tag{22}$$

From this, we define the function class of interest for our setting as

$$\mathcal{F}_{\mathcal{C}_\lambda} := \{f_c(x) = \langle \phi(x), c \rangle : c \in \mathcal{C}_\lambda\} \subset \mathcal{F} = \text{span}\{\psi_1, \psi_2, \ldots, \psi_n\}. \tag{23}$$

Appealing to the boundedness of each fixed basis function, we can also bound the features from the mapping $x \mapsto \phi(x) \in \mathbb{R}^n$ as

$$\sup_{x \in \mathcal{X}} \|\phi(x)\|_2 \leq R\sqrt{n}. \tag{24}$$

We now establish a Lipschitz bound with respect to the first input of the squared-error loss function. Indeed, this loss function is unbounded for *arbitrary* inputs, but the boundedness of $\phi(x)$ and $y$ yields the following bound. Given two inputs $z = f_c(x), z' = f_{c'}(x')$, we have that for a $y \in \mathcal{Y}$

$$|\ell(z, y) - \ell(z', y)| = |(z - y)^2 - (z' - y)^2| \tag{25}$$

$$= |z + z' - 2y||z - z'| \tag{26}$$

$$\leq 2\left(\sup_z |z| + Y\right)|z - z'| \tag{27}$$

$$\leq 2\left(\sup_{x \in \mathcal{X}} \|\phi(x)\|_2 \sup_{c \in \mathcal{C}_\lambda} \|c\|_2 + Y\right)|z - z'| \tag{28}$$

$$\leq 2Y\left(R\sqrt{\frac{n}{\lambda}} + 1\right)|z - z'|. \tag{29}$$

From these bounds, we can apply Theorem 3 from (Kakade et al., 2008) to establish that the Rademacher complexity of $\mathcal{F}_{\mathcal{C}_\lambda}$ can be bounded as

$$\mathcal{R}_m(\mathcal{F}_{\mathcal{C}_\lambda}) = \mathbb{E}\left[\frac{1}{m} \sup_{f \in \mathcal{F}_{\mathcal{C}_\lambda}} \sum_{i=1}^m f_c(x_i)\epsilon_i\right] \leq YR\sqrt{\frac{n}{m\lambda}}, \tag{30}$$

since the regularization function in our case is simply the 2-norm of the coefficients, $c$. Furthermore, this leads to the straightforward application of Rademacher-based generalization bounds for bounded, Lipschitz loss functions, as in Corollary 5 in (Kakade et al., 2008),

$$L(f_{\hat{c}_\lambda}) \leq \hat{L}_m(f_{\hat{c}_\lambda}) + 4\left(R\sqrt{\frac{n}{\lambda}} + 1\right)YR\sqrt{\frac{n}{m\lambda}} + 2\left(R\sqrt{\frac{n}{\lambda}} + 1\right)YR\sqrt{\frac{n\log(1/\delta)}{2m\lambda}} \tag{31}$$

$$= \hat{L}_m(f_{\hat{c}_\lambda}) + 2Y^2 R\sqrt{\frac{n}{m\lambda}}\left(R\sqrt{\frac{n}{\lambda}} + 1\right)\left(2 + \sqrt{\frac{\log(1/\delta)}{2}}\right) \tag{32}$$

$$\leq \hat{L}_m(f_{\hat{c}_\lambda}) + \tilde{\mathcal{O}}\left(Y^2 R^2 \frac{n}{\lambda\sqrt{m}}\right), \tag{33}$$

as desired. $\qquad\square$

The Rademacher complexity of a class of functions indicates the inherent tradeoff between data and the number of basis functions. In our setting, we see that the Rademacher complexity for the relevant function class scales as $\mathcal{R}_m(\mathcal{F}_\lambda) \in \mathcal{O}(\sqrt{n/m\lambda})$. As we increase the number of basis functions $n$, the Rademacher complexity increases, indicating we can model more complex functions, but also risks overfitting unless we have enough data. Conversely, increasing either the sample size $m$ reduces the effective complexity of the class and controls variance.

Statistical consistency requires balancing these two effects. As $m \to \infty$, one must let $\lambda \to 0$ so that the regularizer does not dominate and introduce asymptotic bias. At the same time, $\lambda$ cannot decay too quickly: if $\lambda$ shrinks slower than $1/m$, the bound above ensures that $\mathcal{R}_m(\mathcal{F}_\lambda) \to 0$, guaranteeing vanishing estimation error. This interplay between approximation bias (controlled by $n$ and $\lambda$) and variance (controlled by $m$ and $\lambda$) is exactly the classical bias–variance tradeoff in ridge regression.

## C.2 PROOF OF THEOREM 2

In this section, we provide a proof of Theorem 2, restated below:

**Theorem 2.** *Let $\psi_1, \ldots, \psi_n \subset \mathcal{H}$ be a fixed set of basis functions, where each $\psi_j : \mathcal{X} \to \mathcal{Y}$ is a fixed, bounded neural network satisfying $\sup_{x \in \mathcal{X}} |\psi_j(x)| \leq R$. Assume furthermore that $\mathcal{Y}$ is uniformly bounded as $\sup_{y \in \mathcal{Y}} \|y\|_2 \leq Y$ and that the mapping $\Phi(c) = \sum_{i=1}^n c_i \psi_i$ is injective. Given regularization parameter $\lambda > 0$, then for any $\delta > 0$ we have that with probability greater than or equal to $1 - \delta$ the least-squares solution $f_{\hat{c}_\lambda}$ from (10) satisfies*

$$L(f_{\hat{c}_\lambda}) \lesssim \hat{L}_m(f_{\hat{c}_\lambda}) + \tilde{\mathcal{O}}\left(Y^2 R^2 \frac{n^{3/2}}{\lambda\sqrt{m}}\right). \tag{13}$$

Prior to the proof, we provide a few useful lemmas.

**Lemma 3.** *Given $\eta = \mathcal{N}(a, \alpha^2 I_n)$ and the domain $\mathcal{R} = \{x \in \mathbb{R}^n : \|x - a\|_2 \leq r\} =: B(a, r)$, then the truncated distribution $\eta^{\mathcal{R}}$ of $\eta$ has density $d\eta^{\mathcal{R}}(x) = d\eta(x)/Z$, where*

$$Z = \frac{\gamma(n/2, r^2/2)}{\Gamma(n/2)},$$

*and furthermore we have that*

$$\mathbb{E}_{\eta^{\mathcal{R}}}\left[\|x - a\|_2^2\right] = 2\alpha^2 \frac{\gamma(n/2 + 1, r^2/2)}{\gamma(n/2, r^2/2)}, \tag{34}$$

*where $\Gamma(z) = \int_0^\infty e^{-t} t^{z-1} dt$ is the Gamma function and $\gamma(s, z) = \int_0^z e^{-t} t^{s-1} dt$ is the incomplete gamma function.*

*Proof.* For both desired equalities, we appeal to the fact that the norm squared of a multivariate normal random variable in $\mathbb{R}^n$ follows a chi-squared distribution with $n$ degrees of freedom. From this, we can write the desired integrals in terms of incomplete gamma functions and the standard Gamma function.

First, note that we can apply a change of variables $y = (x - a)/\alpha$ to obtain

$$Z = \int_{\mathcal{R}} d\eta(x) = \int_{\mathcal{R}} \frac{e^{-\|x-a\|_2^2/2\alpha^2}}{(2\pi\alpha^2)^{n/2}} dx = \int_{\|y\|_2 \leq r} \frac{e^{-\|y\|_2^2}2}{(2\pi)^{n/2}} dy \tag{35}$$

$$= \mathbb{P}_{Y \sim \mathcal{N}(0, I_n)}\left(\|Y\|_2 \leq r\right) \tag{36}$$

$$= \mathbb{P}_{C \sim \chi_n^2}\left(C \leq r^2\right) \tag{37}$$

$$= \frac{\gamma(n/2, r^2/2)}{\Gamma(n/2)}. \tag{38}$$

Now, we apply a similar change of variables to obtain

$$\mathbb{E}_{\eta^{\mathcal{R}}}\left[\|x-a\|_2^2\right] = \frac{\alpha^2}{Z}\int_{\|y\|_2\leq r}\|y\|_2^2\frac{e^{-\|y\|^2/2}}{(2\pi)^{n/2}}dy \tag{39}$$

$$= \frac{\alpha^2}{Z}\mathbb{E}_{Y\sim\mathcal{N}(0,\mathrm{I}_n)}\left[\|Y\|_2^2\mathbb{1}\{\|Y\|_2\leq r\}\right] \tag{40}$$

$$= \frac{\alpha^2}{Z}\mathbb{E}_{C\sim\chi_n^2}\left[C\mathbb{1}\{C\leq r^2\}\right] \tag{41}$$

$$= \frac{2\alpha^2\Gamma(n/2)}{\gamma(n/2,r^2/2)}\frac{\gamma(n/2+1,r^2/2)}{\Gamma(n/2)} \tag{42}$$

$$= 2\alpha^2\frac{\gamma(n/2+1,r^2/2)}{\gamma(n/2,r^2/2)}, \tag{43}$$

as desired.

$\square$

**Lemma 4.** *For $s,a > 0$, the incomplete gamma function $\gamma(s,a) = \int_0^a e^{-t}t^{s-1}dt$ satisfies the following bound*

$$\frac{\gamma(s,4a)}{\gamma(s,a)} \leq 4^s. \tag{44}$$

*Proof.* Utilizing a $u$-substitution, we can write

$$\frac{\gamma(s,4a)}{\gamma(s,a)} = \frac{\int_0^{4a}e^{-t}t^{4a-1}dt}{\int_0^a e^{-t}t^{a-1}dt} = \frac{4^s\int_0^u e^{-4u}t^{u-1}du}{\int_0^a e^{-t}t^{a-1}dt} \tag{45}$$

$$\leq 4^s\frac{\int_0^u e^{-u}t^{u-1}du}{\int_0^a e^{-t}t^{a-1}dt} \tag{46}$$

$$= 4^s, \tag{47}$$

where in the second line we have used the fact that $e^{-4u}\leq e^{-u}$. $\square$

Now we turn to the proof of Theorem 2.

*Proof.* We begin by bounding the squared error loss function on an appropriate domain of interest, namely

$$\mathcal{S}_0 := \left\{c\in\mathbb{R}^n : \|c\|_2\leq\frac{Y}{\sqrt{\lambda}}+\sigma\sqrt{n}\right\}, \tag{48}$$

where the constant $\sigma > 0$ will defined hereafter, $n$ is the number of basis functions for the feature $\phi(x)\in\mathbb{R}^n$, $\lambda$ is the regularization parameter, and $Y$ is the uniform bound on outputs, $y\in\mathcal{Y}$. Notice that, given our boundedness assumptions on $\phi(x)$ and $\mathcal{Y}$, we can bound the loss function as

$$\ell(\langle\phi(x),c\rangle,y)\leq\max\left\{\|\phi(x)\|_2\|c\|_2,Y\right\}^2\leq\max\left\{R\sqrt{n}\left(\frac{Y}{\sqrt{\lambda}}+\sigma\sqrt{n}\right),Y\right\}^2 =: A_\sigma \tag{49}$$

for all $c\in\mathcal{S}_0$.

Now, consider the scaled loss function, $\tilde{\ell}(z,y) = \ell(z,y)/A_\sigma$ so that $\tilde{\ell}(z,y)\in[0,1]$. Then, we can apply Corollary 8 from (Kakade et al., 2008) to obtain the following PAC-Bayes bound:

$$\mathbb{E}_{x,y}\left[\mathbb{E}_{f\sim\nu}\left[\tilde{\ell}(f(x),y)\right]\right]\leq\frac{1}{m}\sum_{i=1}^m\mathbb{E}_{f\sim\nu}\left[\tilde{\ell}(f(x_i),y_i)\right]+4.5\sqrt{\frac{\max\{D_{KL}(\nu||\nu_0),2\}}{m}}+\sqrt{\frac{\log(1/\delta)}{2m}}, \tag{50}$$

where $\nu_0$ and $\nu$ are respectively prior and posterior distributions over $f\in\mathcal{F}$.

Let $\Phi(c) = \sum_{i=1}^n c_i\psi_i\in\mathcal{F}\subset\mathcal{H}$ represent the mapping of coefficient in $\mathbb{R}^n$ to functions in $\mathcal{H}$. Furthermore, introduce the domain

$$\mathcal{S} := \{c\in\mathcal{S} : \|c-\hat{c}_\lambda\|_2\leq\sigma\sqrt{n}\}, \tag{51}$$

and we then define the following distributions:

- $\mu_0^{\mathcal{S}_0} = \mathcal{N}_{\mathcal{S}_0}(0, \sigma_0^2 I_n)$, an isotropic, mean-zero multivariate Gaussian truncated to the domain $\mathcal{S}_0$.

- $\mu_0 = \mathcal{N}(0, \sigma_0^2 I_n)$, the *untruncated* counterpart of $\mu_0^{\mathcal{S}_0}$ defined above. That is, the density $d\mu_0^{\mathcal{S}_0}(x) = d\mu_0(x)/Z_0$ for some normalization constant $Z_0 > 0$.

- $\mu^{\mathcal{S}} = \mathcal{N}_{\mathcal{S}}(\hat{c}_\lambda, \sigma^2 I_n)$, an isotropic multivariate Gaussian truncated to the domain $\mathcal{S} \subset \mathcal{S}_0$.

- $\mu = \mathcal{N}(\hat{c}_\lambda, \sigma^2 I_n)$, the *untruncated* counterpart of $\mu^{\mathcal{S}}$ defined above. That is, the density $d\mu^{\mathcal{S}}(x) = d\mu(x)/Z$ for some normalization constant $Z > 0$.

We define the prior $\nu_0$ to be the push-forward of the truncated multivariate Gaussian, $\mu_0^{\mathcal{S}}$, under the mapping $\Phi$

$$f \sim \nu_0 = \Phi_\# \mu_0^{\mathcal{S}}. \tag{52}$$

Similarly, we define the posterior $\nu$ as:

$$f \sim \nu = \Phi_\# \mu^{\mathcal{S}}. \tag{53}$$

We choose $\sigma_0, \sigma$ in what follows to simplify the terms of the PAC-Bayes bound in (50).

Due to the convenient form of the $\nu$ as the push forward of a multivariate Gaussian truncated to a ball centered at $\hat{c}_\lambda$, the expectations with respect to $\nu$ of the squared error loss in (50) can be replaced by the mean of $\nu$, $f = \Phi(\hat{c}_\lambda)$:

$$\mathbb{E}_{x,y}\Big[\tilde{\ell}(f(x), y)\Big] \le \frac{1}{m} \sum_{i=1}^m \tilde{\ell}(f(x_i), y_i) + 4.5 \sqrt{\frac{\max\{D_{KL}(\nu||\nu_0), 2\}}{m}} + \sqrt{\frac{\log(1/\delta)}{2m}}. \tag{54}$$

Furthermore, due to the injectivity of the mapping $\Phi : \mathbb{R}^n \to \mathcal{F}$, the KL divergence between the posterior, $\nu$, and prior, $\nu_0$, can be replaced by the KL divergence between $\mu^{\mathcal{S}}$ and $\mu_0^{\mathcal{S}_0}$; that is,

$$D_{KL}(\nu||\nu_0) = D_{KL}(\Phi_\# \mu^{\mathcal{S}} || \Phi_\# \mu_0^{\mathcal{S}_0}) = D_{KL}(\mu^{\mathcal{S}} || \mu_0^{\mathcal{S}_0}). \tag{55}$$

Now, since $\mathcal{S} \subset \mathcal{S}_0$, the KL divergence between these truncated, multivariate Gaussians can be computed as follows:

$$D_{KL}(\mu^{\mathcal{S}} || \mu_0^{\mathcal{S}_0}) = \mathbb{E}_{\mu^{\mathcal{S}}} \left[ \log \left( \frac{Z_0}{Z} \left( \frac{\sigma_0^2}{\sigma^2} \right)^{n/2} \exp \left( \frac{\|x\|^2}{2\sigma_0^2} - \frac{\|x - \hat{c}_\lambda\|^2}{2\sigma^2} \right) \right) \right] \tag{56}$$

$$= \log \left( \frac{Z_0}{Z} \right) + \frac{n}{2} \log \left( \frac{\sigma_0^2}{\sigma^2} \right) + \frac{1}{2\sigma_0^2} \underbrace{\mathbb{E}_{\mu^{\mathcal{S}}} \left[ \|x\|^2 \right]}_{(I)} - \frac{1}{2\sigma^2} \underbrace{\mathbb{E}_{\mu^{\mathcal{S}}} \left[ \|x - \hat{c}_\lambda\|^2 \right]}_{(II)}. \tag{57}$$

Note that (I) can be written as follows

$$\mathbb{E}_{\mu^{\mathcal{S}}} \left[ \|x\|^2 \right] = \mathbb{E}_{\mu^{\mathcal{S}}} \left[ \|x - \hat{c}_\lambda + \hat{c}_\lambda\|^2 \right] \tag{58}$$

$$= (II) + \|\hat{c}_\lambda\|^2 + 2\hat{c}_\lambda^\top \mathbb{E}_{\mu^{\mathcal{S}}} \left[ x - \hat{c}_\lambda \right] \tag{59}$$

$$= (II) + \|\hat{c}_\lambda\|^2, \tag{60}$$

so we can write (57) as

$$D_{KL}(\mu^{\mathcal{S}} || \mu_0^{\mathcal{S}_0}) = \log \left( \frac{Z_0}{Z} \right) + \frac{n}{2} \log \left( \frac{\sigma_0^2}{\sigma^2} \right) + \frac{\|\hat{c}_\lambda\|^2}{2\sigma_0^2} + \frac{\sigma^2 - \sigma_0^2}{2\sigma_0^2 \sigma^2} \mathbb{E}_{\mu^{\mathcal{S}}} \left[ \|x - \hat{c}_\lambda\|^2 \right]. \tag{61}$$

Applying Lemma 3, we choose $\sigma = Y/\sqrt{n\lambda}$ from which we can simplify and bound the ratio

$$\frac{Z_0}{Z} = \frac{\gamma \left( \frac{n}{2}, \frac{1}{2} \left( \frac{Y}{\sqrt{\lambda}} + \sigma \sqrt{n} \right)^2 \right)}{\gamma \left( \frac{n}{2}, \frac{\sigma^2 n}{2} \right)} = \frac{\gamma \left( \frac{n}{2}, \frac{2Y^2}{\lambda} \right)}{\gamma \left( \frac{n}{2}, \frac{Y^2}{2\lambda} \right)} \le 2^n, \tag{62}$$

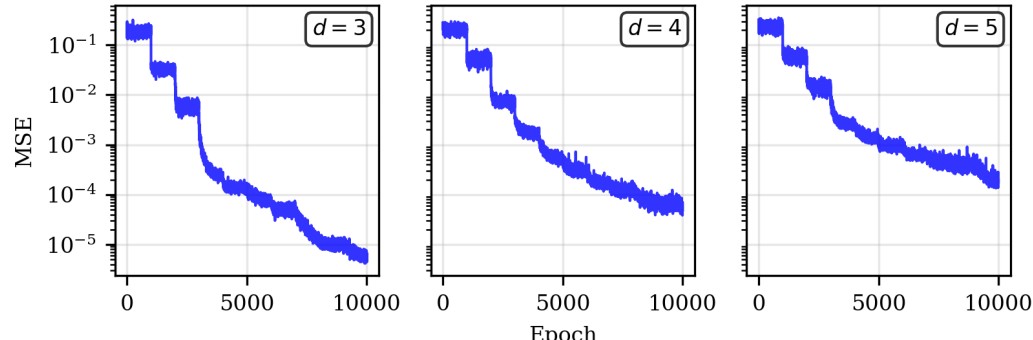

Figure 4: Mean squared error of the progressive training algorithm. We see a plateau in the MSE reduction after the number of basis functions matches the intrinsic dimension of the data.

where in the last inequality we have used Lemma 4. Finally, we choose $\sigma_0^2 = 4\sigma^2$, and recalling that $\|\hat{c}_\lambda\|_2 \leq Y/\sqrt{\lambda}$, we can use Lemma 3 again to bound

$$D_{KL}(\mu^{\mathcal{S}}\|\mu_0^{\mathcal{S}_0}) \leq n\log 2 + \frac{n}{2}\log(4) + \frac{\|\hat{c}_\lambda\|_2^2}{8\sigma^2} + \frac{-3\sigma^2}{8\sigma^4}\mathbb{E}_{\mu^{\mathcal{S}}}\left[\|x - \hat{c}_\lambda\|^2\right] \tag{63}$$

$$\leq n\log 2 + \frac{n}{2}\log(4) + \frac{n}{8} - \frac{3}{4}\frac{\gamma(n/2+1, \sigma^2 n/2)}{\gamma(n/2, \sigma^2 n/2)} \tag{64}$$

$$= n\log 2 + \frac{n}{2}\log(4) + \frac{n}{8} - \frac{3}{4}\frac{\gamma(n/2+1, Y^2/2\lambda)}{\gamma(n/2, Y^2/2\lambda)} \tag{65}$$

$$\leq n\left(2\log 2 + \frac{1}{8}\right) \tag{66}$$

$$\leq 2n. \tag{67}$$

Furthermore, our choice of $\sigma = Y/\sqrt{n\lambda}$ gives that (49) simplifies to $A_\sigma = Y^2\max\{\frac{nR^2}{\lambda}, 1\} = nR^2Y^2/\lambda$. Combining it all together, we can multiply both sides of (54) by $A_\sigma$ to obtain

$$\mathbb{E}_{x,y}\left[\ell(f(x), y)\right] \leq \frac{1}{m}\sum_{i=1}^m \ell(f(x_i), y_i) + A_\sigma\left(4.5\sqrt{\frac{\max\{D_{KL}(\nu\|\nu_0), 2\}}{m}} + \sqrt{\frac{\log(1/\delta)}{2m}}\right) \tag{68}$$

$$\leq \frac{1}{m}\sum_{i=1}^m \ell(f(x_i), y_i) + \frac{nR^2Y^2}{\lambda\sqrt{m}}\left(4.5\sqrt{2n} + \sqrt{\frac{\log(1/\delta)}{2}}\right) \tag{69}$$

$$\lesssim \frac{1}{m}\sum_{i=1}^m \ell(f(x_i), y_i) + \tilde{\mathcal{O}}\left(\frac{R^2Y^2 n^{3/2}}{\lambda\sqrt{m}}\right), \tag{70}$$

as desired. $\qquad\square$

# D   ADDITIONAL RESULTS, EXPERIMENTAL SETUP, AND PARAMETERS

## D.1   POLYNOMIAL REGRESSION

For each degree $d$, we sample random polynomials by drawing coefficients independently and identically distributed from $[-1, 1]$. For each polynomial, we generate 1000 input-output pairs to form a dataset $D_i = \{(x_j, f_i(x_j))\}_{j=1}^{1000}$ (c.f. Ingebrand et al., 2025). We use 100 sampled evaluation points to compute the coefficients at inference time. Each learned basis function is implemented as a one-hidden-layer MLP with width 32.

For the progressive training approach in Algorithm 1, we begin with a single basis function and add new basis functions sequentially. After training each basis function, we compute the coefficient

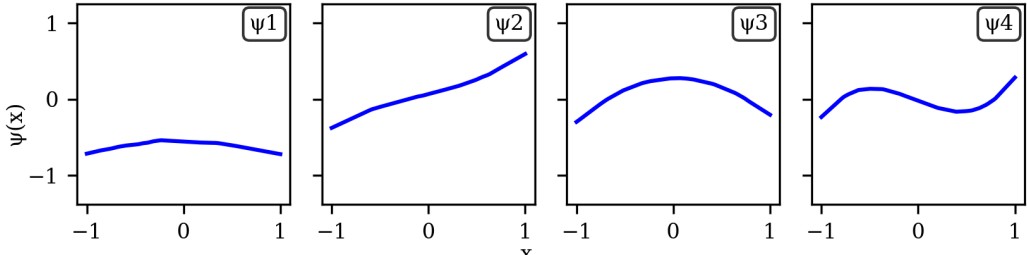

Figure 5: As expected, the progressive approach produces basis functions that mirror the natural ordering found in polynomial basis expansion, where the basis are approximately constant, linear, quadratic, and cubic. This structure emerges because each new basis is trained to capture remaining variance after freezing the previous ones, resulting in interpretable features.

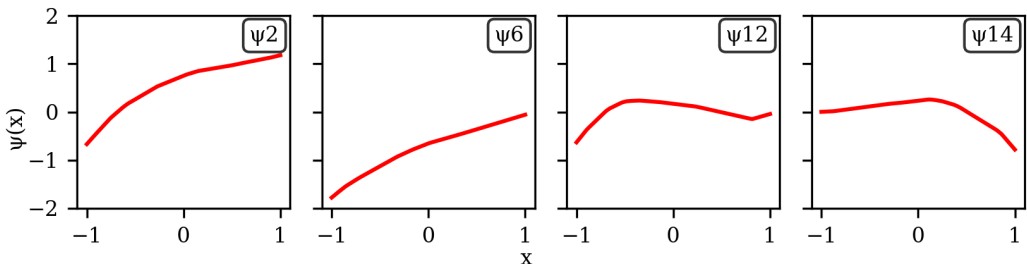

Figure 6: Pruning from an overparameterized encoder yields basis functions with less interpretable structure. Since we trained all bases at the same time, they lack the ordered progression seen in the progressive approach.

matrix $C$ across all datasets, perform PCA, and check the explained variance of the latest component. Training ends once the explained variance drops below a user-specified threshold ($\tau = 1\%$ in our experiments). For the train-then-prune algorithm (Algorithm 2), we begin with an over-specified function encoder with $n = 20$ basis functions. After training, we determine the effective rank using PCA as before, i.e., the smallest $n$ such that the cumulative explained variance is greater than $99\%$. We then select the basis functions corresponding to the principal directions, prune the remaining basis functions, and fine-tune the network for a small number of epochs.

### D.1.1 DEEP KERNEL

Deep kernel learning (DKL) combines neural networks with kernel methods by using a learned feature map $\phi_\theta$ inside a kernel, typically $k_\theta(x, x') = k_{\text{base}}(\phi_\theta(x), \phi_\theta(x'))$ (Wilson et al., 2016). In its standard form, DKL is trained for a single supervised dataset by optimizing the kernel parameters $\theta$ (and possibly kernel hyperparameters) to maximize marginal likelihood in a Gaussian process or to minimize empirical loss in kernel ridge regression. The method is designed to adapt the kernel geometry for a specific task rather than to learn a function space shared across tasks.

Our setting differs: function encoders learn basis functions that span a reusable function class across many training functions. To compare fairly, we adapt deep kernels to the multi-task setting.

Specifically, for each mini-batch of training functions $\{D_j\}$, we construct the evaluation Gram matrix $K_{ij} = k_\theta(x_i, x_j)$, and solve the kernel ridge regression system $(K + \lambda m I)\alpha = y$ for coefficients $\alpha$. Training then backpropagates through the $\mathcal{O}(m^3)$ Gram matrix inversion, updating $\theta$ via gradient descent. This procedure is repeated across functions, treating DKL as if it were learning a space of functions rather than a single-task kernel.

We train the deep kernels and function encoders over the same space of degree-3 polynomials as in Appendix D.1. We separate the data into a set of query points and a set of evaluation points

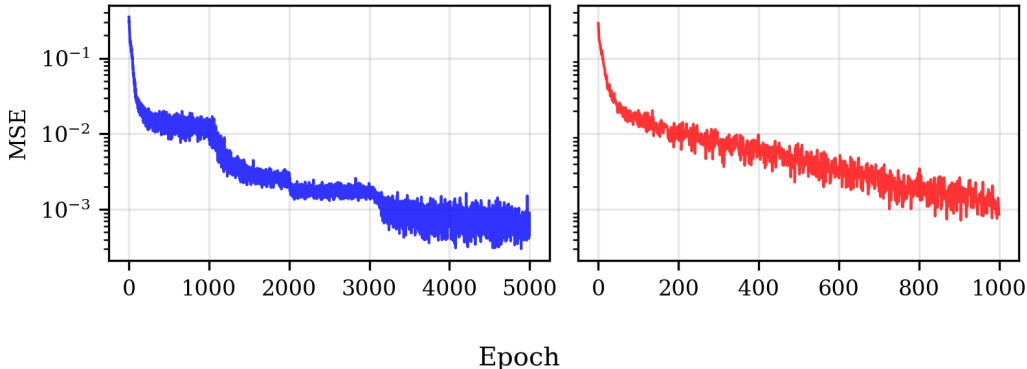

Figure 7: Mean squared error for the progressive training algorithm (left) and the train-then-prune algorithm (right) on the Van der Pol system.

to compute the coefficients. The inputs are mapped through the feature network $\theta$ to produce the evaluation embedding $Z_E \in \mathbb{R}^{m \times d}$ and the query embedding $Z_Q \in \mathbb{R}^{q \times d}$, with $m$ the number of evaluation points, $q$ the number of query points, and $d$ the output dimension of the network.

The first deep kernel is an RBF kernel in feature space with automatic relevance determination lengthscales,

$$k_\theta(x, x') = \sigma^2 \exp\left(-\frac{1}{2} \sum_{j=1}^{d} \frac{(\theta_j(x) - \theta_j(x'))^2}{\ell_j^2}\right) \tag{71}$$

where $\ell \in \mathbb{R}_{>0}^m$ are learnable lengthscales and $\sigma^2 > 0$ is the output scale. These embeddings define Gram matrices $K_{EE} = k_\theta(Z_E, Z_E)$ and $K_{QE} = k_\theta(Z_Q, Z_E)$. A kernel ridge regression predictor is fitted on the example set by solving $\alpha_E = (K_{EE} + \lambda I)^{-1} y_E$, and predictions for the full dataset are given by $\hat{y} = K_{DE}\alpha_E$. The model parameters, including both the feature extractor $\theta$ and the kernel parameters, are optimized by minimizing the mean-squared error on $\hat{y}$ against the full targets $y$, averaged across functions in the batch.

For the second part of the experiments, we replace the RBF kernel with a linear kernel. With the same feature extractor $\theta$ and embeddings $Z_E, Z_Q$ as above, the kernel is

$$k_\theta(x, x') = \theta(x)^\top \theta(x'). \tag{72}$$

This induces the Gram matrices $K_{EE} = Z_E Z_E^\top$ and $K_{QE} = Z_Q Z_E^\top$. The training and prediction follow the same kernel ridge regression procedure described in the previous paragraph

The neural network for the RBF deep kernel is a one-hidden-layer MLP with a width of 64. The function encoder is the same as in D.1. The linear deep kernel uses the same neural network architecture as the function encoder.

### D.2   VAN DER POL OSCILLATOR

We evaluate our method on the Van der Pol oscillator, defined by $\dot{x}_1 = x_2$ and $\dot{x}_2 = \mu(1 - x_1^2)x_2 - x_1$ with $\mu \in [0.5, 2.5]$. Training data are generated by uniformly sampling initial conditions $x_0 \in [-3.5, 3.5]^2$ and integrating trajectories over $t \in [0, 10]$ with time step $\Delta t = 0.1$. We generate a dataset with 1000 query points and 100 evaluation points. Both the progressive training and train-then-prune methods use an MLP with two hidden layers of width 64, mapping inputs $(x_1, x_2, \mu)$ to a two-dimensional output. The progressive training starts with 5 basis functions, whereas train-then-prune starts with 10. The mean squared error is plotted in Figure 7. After training, our algorithms produce accurate models with fewer basis functions. A representative trajectory from each algorithm on the Van der Pol system is shown in Fig. 8.

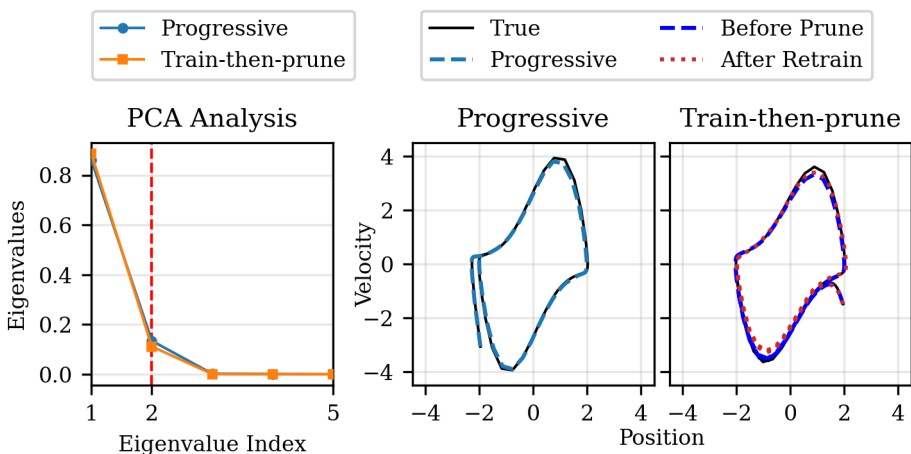

Figure 8: (Left) Eigenvalue spectra of the covariance matrix for both the progressive and train-then-prune approaches reveal a similar trend and identify that only two basis functions are needed to capture nearly all variance. (Middle and Right) Predicted trajectories using the two methods accurately capture the nonlinear oscillatory dynamics using just two basis functions.

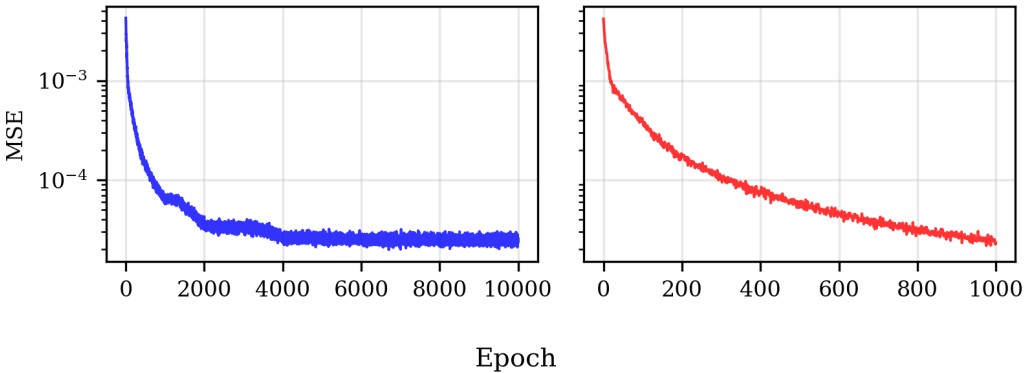

Figure 9: Training loss (MSE) curves for the two-body system. (Left) Progressive training gradually reduces error over multiple stages as each basis function is added and optimized. Unlike the Van der Pol system, the MSE loss shows a more gradual decrease due to the complexity of the system. (Right) Train-then-prune training proceeds with all bases jointly, showing steady but slower convergence.

### D.3   TWO-BODY PROBLEM

We study the planar two-body problem with point-mass dynamics $\ddot{\mathbf{r}} = -\mu\,\mathbf{r}/\|\mathbf{r}\|^3$. We generate initial conditions by sampling Keplerian elements with semi-major axis $a \in [1.0, 3.0]$, eccentricity $e \in [0, 0.7]$ (to avoid singularities), argument of periapsis $\omega \in [0, 2\pi]$, and gravitational parameter $\mu \in [0.8, 1.1]$. The sampled elements are converted to Cartesian states and propagated over $t \in [0, 50]$ with time step $\Delta t = 0.05$. Each trajectory is sampled at $1{,}000$ time points, with $100$ points held out for evaluation. Both training variants share an MLP with two hidden layers of width $64$ and take as input the gravitational parameter together with the Cartesian coordinates generated from the orbital elements; the progressive variant is initialized with $5$ basis functions, while the train-then-prune variant is initialized with $10$ basis functions and pruned afterward.

The mean squared error of the two algorithms is shown in Fig. 9. A representative trajectory from the two algorithms is shown in Fig. 10. Streamplots of the selected basis functions after training are shown in Fig. 11 and Fig. 12.

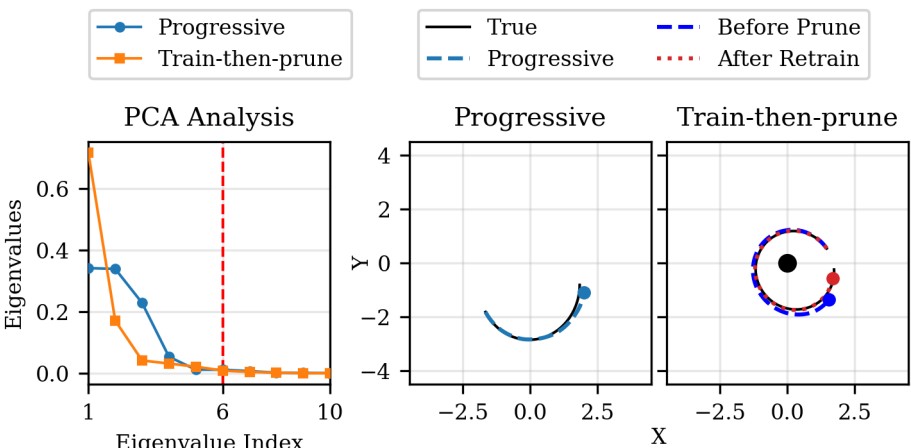

Figure 10: (Left) For the two-body problem, eigenvalue decay is more gradual, reflecting the increased complexity of elliptical orbits. Both methods require 5-6 bases to capture the function space. Unlike the Polynomial and Van der Pol systems, the two approaches show more distinct behavior. Both methods (middle and right) successfully learn a compact function encoder that accurately reproduces elliptical dynamics.

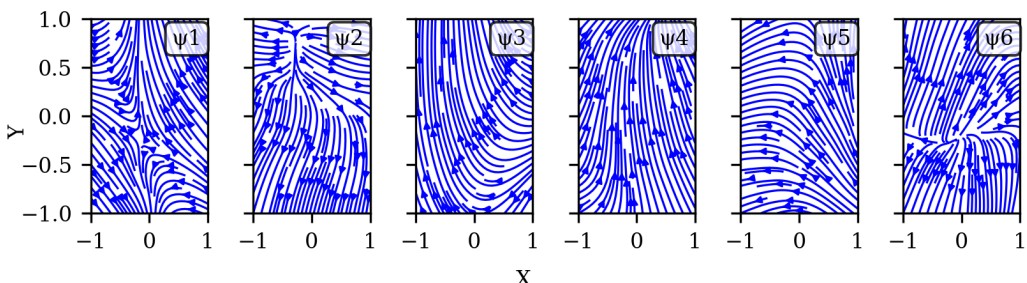

Figure 11: The progressive approach yields basis functions that exhibit more complex patterns across the state space for the two-body problem.

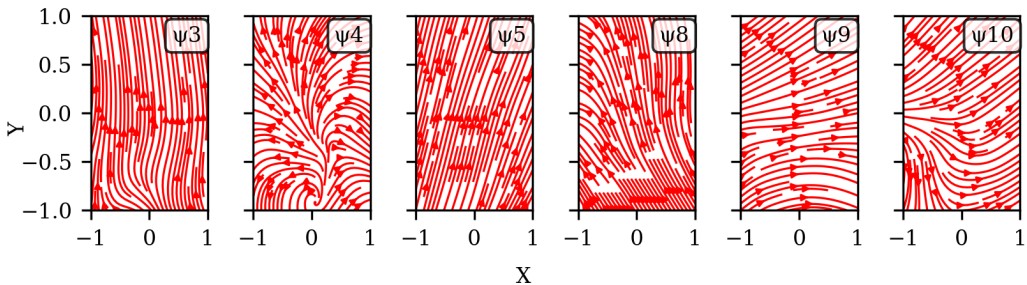

Figure 12: Train-then-prune approach produces more uniform basis functions for the two-body problem.

