# OpenReview forum: "Function Spaces Without Kernels: Learning Compact Hilbert Space Representations"
_ICLR.cc/2026/Conference — ICLR 2026 Conference Desk Rejected Submission_

### Official Review · Reviewer_bCyB · 2025-10-22

**Soundness:** 3
**Presentation:** 2
**Contribution:** 1
**Rating:** 2
**Confidence:** 5

**Summary:**

This paper connects function encoders (neural networks that learn a compact set of basis functions forming an explicit feature map)
$\(\phi(x) = [\psi_1(x), \dots, \psi_n(x)]^\top\)$ to the theory of kernel methods. The inner product $\(\langle \phi(x), \phi(x') \rangle\)$ defines a valid kernel, giving a kernel-theoretic interpretation of learned neural representations. Two PCA-inspired methods (progressive training and train-then-prune) are introduced to identify compact bases, and generalization guarantees are derived using Rademacher complexity and PAC-Bayes analysis. Small-scale experiments on polynomial and dynamical system benchmarks are conducted to study the empirical properties of the approach.

**Strengths:**

The paper is clearly written and reasonably well organized. The exposition is coherent, and the authors succeed in presenting familiar ideas from kernel methods, neural feature learning, and function approximation under a unified formalism.

**Weaknesses:**

In my opinion, the paper adds no real novelty. Its main claims correspond almost exactly to well-established results in the literature, and the work largely repackages known ideas under new terminology. More specifically:

- **On the theoretical contribution:** The statement that function encoders define a kernel through inner products of learned features is a direct restatement of the defining property of reproducing kernel Hilbert spaces, rigorously established since Aronszajn (1950) and detailed in Steinwart and Christmann (2008, Ch. 4). The associated primal–dual correspondence is classical convex duality and has been repeatedly exploited in random features, Nyström approximations, and kernel dictionary learning.

- **On the algorithmic side:** The proposed PCA-based “progressive” and “train-then-prune” methods are simple applications of standard subspace and rank-selection heuristics. Similar procedures have long been employed in representation learning, linear probing of pre-trained networks, and dictionary learning methods, such as K-SVD or matching pursuit.

- **On the theoretical analysis:** The Rademacher and PAC-Bayes bounds are restatements of known results for ridge regression with fixed features. The analysis treats the learned feature map as fixed, thereby sidestepping the genuine statistical challenge of analyzing learned representations.

Finally, the empirical evaluation is limited to small, low-dimensional systems and does not substantiate claims of scalability or generality. I also find that this paper overlooks many classical results from the kernel and approximation literature, while relying heavily on recent “Ingebrand et al.” papers, which provide a narrow and incomplete view of prior work.

**Questions:**

None

---

> ### Author Response · Authors · 2025-11-20
>
> 1. On the theoretical contribution: The statement that function encoders define a kernel through inner products of learned features is a direct restatement of the defining property of reproducing kernel Hilbert spaces, rigorously established since Aronszajn (1950) and detailed in Steinwart and Christmann (2008, Ch. 4). The associated primal–dual correspondence is classical convex duality and has been repeatedly exploited in random features, Nyström approximations, and kernel dictionary learning.
>
> We agree that the observation that a finite feature map induces a kernel through $k(x,x') = \langle \phi(x), \phi(x') \rangle$ is classical. This correspondence is not the contribution. What is new in our work is that the feature map $\phi(x)$ is learned directly from data so that the induced kernel and the associated Hilbert subspace are obtained as a consequence of feature learning, not specified in advance. In contrast to random features, Nyström methods, or kernel dictionaries, which begin with a chosen kernel and operate within its fixed RKHS, our method learns the basis functions $\psi_j$ themselves. The kernel is produced "for free" by the learned features, enabling compact, data-adaptive representations without prespecifying a kernel.
>
> 2. On the algorithmic side: The proposed PCA-based “progressive” and “train-then-prune” methods are simple applications of standard subspace and rank-selection heuristics. Similar procedures have long been employed in representation learning, linear probing of pre-trained networks, and dictionary learning methods, such as K-SVD or matching pursuit.
>
> K-SVD applies SVD to residual matrices to update individual dictionary atoms. These residual matrices isolate the approximation error specific to a single atom and do not represent the joint behavior of the coefficients across the dataset. Consequently, their singular values only reflect local reconstruction error. In our method, PCA is applied to the coefficient covariance across tasks.  This operator directly captures how the learned basis is used, and its eigenvalue spectrum quantifies the number of meaningful directions in the representation.
>
> Matching pursuit is similar to our progressive training in that both add basis elements by fitting residuals. However, matching pursuit selects atoms solely according to their instantaneous correlation with the current residual. It never observes how the basis is used across different tasks, since it does not compute or even access the coefficient vectors for those tasks. Consequently, matching pursuit cannot form the coefficient covariance and therefore cannot recover any meaningful notion of intrinsic dimensionality. In contrast, our progressive training produces a shared learned basis, and the coefficients obtained on multiple tasks yield a covariance operator whose eigenvalue spectrum reveals the effective dimension of the Hilbert subspace.
>
> 3. On the theoretical analysis: The Rademacher and PAC-Bayes bounds are restatements of known results for ridge regression with fixed features. The analysis treats the learned feature map as fixed, thereby sidestepping the genuine statistical challenge of analyzing learned representations.
>
> Our goal in this work is not to introduce new generalization bounds, but to demonstrate that existing bounds already apply to the function-encoder setting and provide a link between the learned basis and generalization. We view this as an initial step: establishing that standard bounds are compatible with function encoder lays the groundwork for developing sharper, more tailored bounds in future work.
>
> 4. Finally, the empirical evaluation is limited to small, low-dimensional systems and does not substantiate claims of scalability or generality.
>
> We will clarify that our notion of scalability refers to not scaling with the number of data points. Because inference is performed in an $n$-dimensional primal space, the computational cost depends on the number of learned basis functions $n$ rather than on the dataset size $m$.
>
> 5. I also find that this paper overlooks many classical results from the kernel and approximation literature, while relying heavily on recent “Ingebrand et al.” papers, which provide a narrow and incomplete view of prior work.
>
> In the revision, we will expand the discussion of related methods to include matching pursuit, and K-SVD, and we will clarify the distinctions from random features, dictionary learning approaches, and Nyström approximations.

---

### Official Review · Reviewer_Yvv4 · 2025-10-24

**Soundness:** 3
**Presentation:** 3
**Contribution:** 3
**Rating:** 6
**Confidence:** 3

**Summary:**

The authors show a connection to feature learning and kernel methods and propose two practical algorithms to learn a compact basis, progressive training and train-than-prune. They also provide theoretical analysis regarding generalization bound based on the Rademacher complexity and PAC-Bayes.

**Strengths:**

The authors investigate learning a compact basis for feature learning from the perspective of both practical algorithms and theory. They provide algorithms to obtain a compact basis and show how the number of basis functions depends on the generalization bound. The topic is interesting and relevant to the community.

**Weaknesses:**

Although both the practical algorithms and the theoretical analysis are interesting, I have the following concerns and questions.
- The connection between the proposed practical algorithms in Section 4 and the theoretical analysis in Section 5 is unclear. For example, the proposed algorithms try to obtain basis functions which span the intrinsic dimension of data by computing the cumulative explained variance CEV. However, the setting of the theoretical analysis is general and how the practical algorithms affect to the generalization bound is not clear. Showing the dependency of the bound on CEV would be interesting.

- In Section 5, the authors derive two bounds (Theorems 1 and 2). The first one depends on $n$ linearly, but the second one depends on $n^{3/2}$. Does this mean the second one is not tight? Why this difference appears?

**Questions:**

Please see the weakness part.

---

> ### Author Response · Authors · 2025-11-20
>
> 1. The connection between the proposed practical algorithms in Section 4 and the theoretical analysis in Section 5 is unclear. For example, the proposed algorithms try to obtain basis functions which span the intrinsic dimension of data by computing the cumulative explained variance CEV. However, the setting of the theoretical analysis is general and how the practical algorithms affect to the generalization bound is not clear. Showing the dependency of the bound on CEV would be interesting.
>
> Sections 4 and 5 are connected through their common dependence on the number of basis functions $n$. The generalization bounds in Section 5 hold for any finite-dimensional feature map and therefore apply directly once the progressive training procedure determines $n$. While our analysis does not explicitly incorporate the CEV, the CEV-based stopping rule in Section 4 is precisely the mechanism that selects $n$ in practice. We agree that incorporating the CEV more explicitly into the theoretical analysis would strengthen the connection between Sections 4 and 5. Doing so would require linking the empirical coefficient spectrum to approximation error, which is an interesting extension of our framework but lies beyond the scope of the current work. We will clarify this relationship in the revision.
>
> 2. In Section 5, the authors derive two bounds (Theorems 1 and 2). The first one depends on $n$ linearly, but the second one depends on $n^{3/2}$. Does this mean the second one is not tight? Why this difference appears?
>
> We address this point in more detail in our response to Reviewer nAUD, where we explain why the PAC-Bayes bound is looser and how, with more informative priors tailored to a given problem, it can be tightened. We refer the reviewer to that discussion for further details.

---

> > ### Comment · Reviewer_Yvv4 · 2025-11-27
> >
> > Thank you for the rebuttal. Although both the practical algorithms and the theoretical analysis in this work would give us some insights, I think the theoretical part should depend more on the proposed algorithm to show advantages of the algorithm. I will keep my score.

---

### Official Review · Reviewer_nAUD · 2025-10-25

**Soundness:** 1
**Presentation:** 3
**Contribution:** 2
**Rating:** 2
**Confidence:** 4

**Summary:**

This paper studies function encoders for regression problems. These are neural network architectures with a nonstandard training strategy that splits the training of the last layer coefficients and basis functions separately. Due to the finite number of bases, there is a connection to a data-adapted finite-dimensional RKHS spaces. Two additional training variants are proposed, one based on greedy PCA and another on fine-tuning. The authors develop generalization gap bounds for fixed basis functions. Numerical results for polynomial approximation and dynamical systems learning support the ideas.

**Strengths:**

The paper is generally well written and clear. The greedy basis-by-basis residual training introduced in Sec. 4 is interesting, though I am not sure how original this is. See for instance the paper Cyr et al, which does something similar. I like the cleanness of the polynomial experiment, especially Fig 5 in the SM. The paper centers on the timely topic of connecting neural networks (less theory) with kernels (more theory), which is important for the field.


[REF] Robust training and initialization of deep neural networks: An adaptive basis viewpoint. Cyr, Eric C and Gulian, Mamikon A and Patel, Ravi G and Perego, Mauro and Trask, Nathaniel A. Mathematical and Scientific Machine Learning. 2020, PMLR.

**Weaknesses:**

The paper does not do a good job of distinguishing training functional encoders from the underlying architecture. The architecture is the set
\begin{align}
	F_n=\\{ f=\sum\_{j=1}^nc\_j\psi\_j: c\in\mathbb{R}^n, \psi\_j\in B \\}
\end{align}
where $B$ is some set of functions. But this is equivalent to a space of neural networks with fixed width $ n $. What distinguishes a functional encoder from a neural network is how it is trained, but fundamentally it is just a neural network architecture.

Contribution 1 (connecting function encoders to feature learning and kernel methods) is not novel. It is well known that the finite-dimensional feature map $ \phi $ defines a finite rank kernel. The literature on Nystrom and random feature approximations of kernel methods exploits this fact, see REFS 1 and 2.

Contribution 3 is also an easy application of known (loose) generalization bounds. Although the authors claim they extend kernel-style analysis to neural predictors, the neural part is frozen and fixed, so it is exactly a kernel analysis that does not take into account any of the neural structure. The actually interesting neural contribution to the error is the empirical risk $\hat{L}\_m(f\_{\hat{c}\_\lambda})$, which is left unquantified as a function of $ n,m,N,\lambda $. Furthermore, Eqn 21 is well known in learning theory to be loose, e.g., if the true function is in the RKHS then $\\|\hat{c}\_\lambda\\|=O(1)$ instead of $O(\lambda^{-1/2})$.

Thus, the main contributions that I see are Contribution 2 (training strategies) and the numerical part.

The paper also downplays the potential of kernel methods. For example, lines 79--85 says ``[kernel or approximate kernel methods] cannot adapt to data.'' But this is not true in practice. If one goes into scikitlearn and trains a kernel regressor on a dataset, it will automatically perform maximum likelihood estimation to adapt the hyperparameters of the kernel to the dataset. This data adapted kernel is then used to perform regression. There is also analysis of such data-adapted kernels, see REFs 3 and 4.



1. Generalization properties of learning with random features. Rudi, Alessandro and Rosasco, Lorenzo. Advances in neural information processing systems, 2017.

2. Less is more: Nystrom computational regularization. Rudi, Alessandro and Camoriano, Raffaello and Rosasco, Lorenzo. Advances in neural information processing systems, 2015.

3. Maximum likelihood estimation and uncertainty quantification for Gaussian process approximation of deterministic functions. Karvonen, Toni and Wynne, George and Tronarp, Filip and Oates, Chris and Sarkka, Simo. SIAM/ASA Journal on Uncertainty Quantification, 2020

4. Convergence of Gaussian process regression with estimated hyper-parameters and applications in Bayesian inverse problems. Teckentrup, Aretha L. SIAM/ASA Journal on Uncertainty Quantification. 2020.

**Questions:**

Questions:

1. What is the role of the set of datasets $\\{D_j\\}_{j=1}^N$? Do you typically view $ N=1 $ or much larger? Similarly, does Algorithm 1 even make sense in the common setting $ N=1 $? Because in that case Eqn 7 is rank one so $ CEV_r=1 $ always in this case and hence the method would stop with only one basis function.
1. The training of $f\in F_n$ is totally unclear from Eqn 4 line 128. What regularization norm appears in Eqn 4?  What is $f_j$ and $\hat{f}_j$, and what spaces do they belong to? How is $\hat{f}_j$ related to the function encoder hypothesis set $F_n$? Is $\hat{f}_j\in F_n$? If so, are the coefficients $c_j$ also learned jointly with the $\psi_j$ in the ``offline'' step?
1. What is the role of the offline step vs. the least squares step Eqns 2--3? Why not just stop at Eqn 4?
1. I feel that the number of tasks $ N $ should appear somewhere in the analysis, especially with the PCA-type training given eqn 7. Why doesn't it?
1. Comparing Eqn 12 and 13 shows that the PAC-Bayes approach gives a strictly worse upper bound; so why even state the result 13? The theorem hypotheses of Thm 1 and 2 appear to be the same.
1. The numerical experiments do not report $ N $, the number of task datatsets for offline training.
1. The authors claim that increasing the regularization parameter ``makes the model less prone to overfitting and improves generalization performance'' in line 683--684 does not consider that the empirical risk can (and will) blow up as $ \lambda \to\infty $. Consistency requires $ \lambda\to 0 $, which again shows that the Rademacher bound is too loose.


Typos:
1. line 41: ``in the primal'' what? The sentence just ends.
1. \item line 133 sentence seems incomplete. And if the basis functions are not unique, then how can they be linearly independent as stated in line 134?

---

> ### Author Response · Authors · 2025-11-20
>
> 1. The greedy basis-by-basis residual training introduced in Sec. 4 is interesting, though I am not sure how original this is. See for instance the paper Cyr et al, which does something similar.
>
> Cyr et al. optimize the hidden representation of a single supervised predictor, where the final layer weights serve as a latent basis but are never required to span a multi-task function class. They interpret the last layer of a deep neural network as a set of basis functions. In contrast, our approach learns an explicit, fixed set of basis functions that define a finite-dimensional Hilbert subspace with an induced kernel. In other words, function encoders learn a shared, finite-dimensional RKHS optimized across many distinct functions, not a single predictor.
>
> 2. The paper does not do a good job of distinguishing training functional encoders from the underlying architecture. The architecture is the set $F_n = \{ f = \sum_{j=1}^n c_j \psi_j : c \in \mathbb{R}^n\; \psi_j \in B \}$ where $B$ is some set of functions. But this is equivalent to a space of neural networks with fixed width $n$. What distinguishes a functional encoder from a neural network is how it is trained, but fundamentally it is just a neural network architecture.
>
> A neural network learns a function, whereas a function encoder learns a function space that defines an explicit learned feature map $\phi(x)$ and an associated kernel.
> Generally, a standard neural network is trained on a single dataset to approximate a single function. Function encoders are trained on a collection of functions.
>
> For each function, the coefficients are solved by regularized least squares, and the encoder backpropagates through these closed-form solutions. This induced RKHS enables kernel-style generalization bounds (Rademacher, PAC-Bayes), which do not apply to standard neural network predictors. Thus, while the parametrization uses neural networks, the object being learned is a Hilbert space representation, not a single neural model.
>
> 3. Contribution 1 (connecting function encoders to feature learning and kernel methods) is not novel. It is well known that the finite-dimensional feature map $\phi$ defines a finite rank kernel. The literature on Nystrom and random feature approximations of kernel methods exploits this fact, see REFS 1 and 2.
>
> We agree that the statement "finite feature maps induce kernels" is classical. Our contribution is that we identify the function encoder as a specific learned finite-dimensional RKHS and characterize the associated (scalar and operator-valued) kernels that arise from the multi-task basis-learning objective. Nyström and RFF require users to predefine a fixed kernel k. Deep kernel learning defines $k(f_\theta(x),f_\theta(x’))$ where the RKHS is that of the chosen base kernel. In contrast, we show that we learn the RKHS itself, not a parameterization inside a pre-specified infinite RKHS.
>
> 4. Contribution 3 is also an easy application of known (loose) generalization bounds. Although the authors claim they extend kernel-style analysis to neural predictors, the neural part is frozen and fixed, so it is exactly a kernel analysis that does not take into account any of the neural structure.
>
> Our contribution is to establish that function encoders admit kernel-style inference-time guarantees, which were not previously available for neural basis models. The bounds are not simply kernel bounds: they show that function encoders, which are neural models, admit kernel-style inference-time guarantees. This is not trivial and establishing that the resulting predictor implicitly defines a kernel once the basis is fixed is a key structural insight. We agree that Eqn. 21 is loose; this is expected for distribution-free worst-case analysis. Tighter, data-dependent PAC-Bayes priors (e.g., centered at task coefficients or aligned with PCA components) would yield sharper guarantees, and Theorem 2 establishes the groundwork for such analysis. Our contribution is to establish the necessary preliminary results: function encoders inherit kernel-style generalization behavior once the basis is fixed, and reducing the number of basis functions improves sample complexity.

---

> ### Author Response · Authors · 2025-11-20
>
> 5. The paper also downplays the potential of kernel methods. For example, lines 79--85 says ``[kernel or approximate kernel methods] cannot adapt to data.'' But this is not true in practice. If one goes into scikitlearn and trains a kernel regressor on a dataset, it will automatically perform maximum likelihood estimation to adapt the hyperparameters of the kernel to the dataset. This data adapted kernel is then used to perform regression. There is also analysis of such data-adapted kernels, see REFS 3 and 4.
>
> We agree that hyperparameter tuning in kernel methods constitutes a form of adaptation. Our claim is specifically that classical kernel methods do not learn the feature map itself: adaptation occurs only within a fixed kernel family, and the RKHS is determined by that choice. Function encoders, in contrast, learn an explicit set of basis functions across multiple tasks, thereby constructing a finite-dimensional representation of the function space without assuming a kernel a priori. Deep kernel learning is related in that it learns a kernel through a neural embedding, but it still generally relies on a base kernel and RKHS and requires $m \times m$ Gram matrices for inference. The cited GP consistency results concern hyperparameter estimation within a predefined kernel class and do not involve learning new basis functions or adapting the RKHS span.
>
> 6. What is the role of the set of datasets $\{D_j\}_{j=1}^N$? Do you typically view $N = 1$ or much larger? Similarly, does Algorithm 1 even make sense in the common setting $N = 1$? Because in that case Eqn 7 is rank one so $C E_V = 1$ always in this case and hence the method would stop with only one basis function.
>
> $N$ denotes the number of distinct functions, typically ranging from hundreds to thousands, not the number of samples from a single function. Each dataset $D_j$ corresponds to one function $f_j$. You are correct that when $N = 1$, the method reduces to standard regression. This is expected, since with only a single function, there is no variation across tasks from which to learn a reusable basis, and the covariance operator becomes uninformative. In that setting, learning a shared feature space is neither necessary nor meaningful, and training a single neural network is the appropriate approach. Our method is explicitly designed for the multi-function regime ($N \gg 1$), where learning a compact Hilbert subspace provides a clear advantage. When $N \gg 1$, the covariance matrix $\Sigma_b$ reflects how the learned basis is used across many distinct functions. In this regime, its eigenvalue spectrum exhibits a clear elbow that corresponds to the intrinsic dimensionality of the function family, allowing us to determine an effective basis size. This behavior is consistently observed in our experiments.
>
> 7. The training of $f \in F_n$ is totally unclear from Eqn 4 line 128. What regularization norm appears in Eqn 4? What is $f_j$ and $\tilde{f}_j$, and what spaces do they belong to? How is $\tilde{f}_j$ related to the function encoder hypothesis set $F_n$? If so, are the coefficients $c_j$ also learned jointly with the $\psi_j$ in the ``offline'' step?
>
> We appreciate the opportunity to clarify the objects appearing in Eqn. 4. Each dataset $D_j$ corresponds to a ground-truth function $f_j\in H$. The function encoder produces an approximation $\tilde f_j\in F_n$ using basis functions $\{\psi_k\}$. The coefficients $c_j$ are not learned jointly with the network parameters. Instead, they are computed at inference time in closed form by ridge regression (Eqn. 3). The regularization in Eqn. 4 is the ridge penalty $\lambda\|c_j\|^2$. Eqn. 4 trains only the basis functions $\psi_k$; $\tilde f_j$ is simply the projection of $f_j$ into the learned hypothesis space $F_n$.
>
> We will clarify this distinction in the paper, along with a more detailed explanation of function encoder training.
>
> 8. What is the role of the offline step vs. the least squares step Eqns 2--3? Why not just stop at Eqn 4?
>
> Offline training (Eqn. 4) learns the basis functions; Least-squares at inference (Eqns. 2–3) computes task-specific coefficients c$\psi_1,\dots,\psi_n$. Stopping at Eqn. 4 would produce basis functions $\psi_k$, but no mechanism for making predictions on a new function. This is not solved during the offline step because: the new function was not available during offline training, its outputs $y_i$ differ from those of the training functions, the optimal coefficients $c$ are task-specific, not shared.

---

> ### Author Response · Authors · 2025-11-20
>
> 9. I feel that the number of tasks $N$ should appear somewhere in the analysis, especially with the PCA-type training given eqn 7. Why doesn't it?
>
> $N$ affects the basis-learning stage, not the inference stage. It is crucial during offline stage because it determines how well we can learn basis functions that span the subspace supported by the data and estimate the covariance in Eqn. (7).
>
> The generalization bounds we present in the paper are inference-time bounds, meaning the ridge regression is analyzed after the basis is fixed, and the hypothesis class does not depend on $N$. A joint analysis of basis-learning and inference would require assumptions on neural approximation properties and is an open problem.
>
> 10. Comparing Eqn 12 and 13 shows that the PAC-Bayes approach gives a strictly worse upper bound; so why even state the result 13? The theorem hypotheses of Thm 1 and 2 appear to be the same.
>
> We include the PAC-Bayes bound because it provides a distribution-dependent framework that can yield tighter guarantees with informative priors (e.g., PCA-aligned or coefficient-centered), whereas Rademacher bounds are distribution-free and cannot exploit such structure. The PAC-Bayes bound is looser in our current presentation because we use a very general, uninformative prior (a truncated Gaussian chosen only to handle unbounded losses). A more informative prior, e.g. centered on the empirical coefficients or exploiting the PCA structure would yield a much tighter bound, often sharper than the distribution-free Rademacher bound. We include Theorem 2 because it establishes this framework for future work. In the paper, we will clarify that the two bounds cover complementary aspects and also clarify the distribution-dependence of Theorem 2.
>
> 11. The numerical experiments do not report $N$, the number of task datasets for offline training.
>
> We used N = 1000 tasks in all three benchmarks; sampling details appear in Appendix D. We will move N and the key sampling ranges into the main text for clarity.
>
> 12. The authors claim that increasing the regularization parameter ``makes the model less prone to overfitting and improves generalization performance'' in line 683--684 does not consider that the empirical risk can (and will) blow up as $\lambda \to \infty$. Consistency requires $\lambda \to 0$, which again shows that the Rademacher bound is too loose.
>
> We agree. The earlier statement focused only on the estimation term in the bound. We now clarify the bias–variance tradeoff and the requirement that $\lambda \to 0$ as $m \to \infty$ for consistency and have removed the misleading sentence.
>
> Typos:
>
> 1. line 41: ``in the primal'' what? The sentence just ends.
>
> We now write "in the primal space."
>
> 2. line 133 sentence seems incomplete. And if the basis functions are not unique, then how can they be linearly independent as stated in line 134?
>
> We clarify that basis functions are distinct and linearly independent, but the learned basis is not unique across training runs.

---

> ### Comment · Reviewer_nAUD · 2025-11-26
>
> I thank the authors for their detailed reply to my review. The answers to my questions has helped improve my understanding of the paper. However, my criticisms in the "Weaknesses" section remain and are shared with the criticism made by reviewer bCyB who I completely agree with.
>
> The authors emphasize the feature map learning without relying on a base kernel is what function encoders do, which I agree with. One can then use the function encoder basis to regress a fixed function at inference time with the basis frozen. However, there are similar works along these lines that also do not rely on base kernels (and work with finite-dim. feature spaces like this paper):
>
> - Implicit kernel meta-learning using kernel integral forms. John Isak Texas Falk, Carlo Cilibert, Massimiliano Pontil Proceedings of the Thirty-Eighth Conference on Uncertainty in Artificial Intelligence, PMLR 180:652-662, 2022.
>
> - Implicit Kernel Learning. Chun-Liang Li, Wei-Cheng Chang, Youssef Mroueh, Yiming Yang, Barnabas Poczos Proceedings of the Twenty-Second International Conference on Artificial Intelligence and Statistics, PMLR 89:2007-2016, 2019.
>
> - @article{yoo2021deep,
>   title={Deep regularization and direct training of the inner layers of neural networks with kernel flows},
>   author={Yoo, Gene Ryan and Owhadi, Houman},
>   journal={Physica D: Nonlinear Phenomena},
>   volume={426},
>   pages={132952},
>   year={2021},
>   publisher={Elsevier}
> }
>
> - @article{owhadi2019kernel,
>   title={Kernel flows: From learning kernels from data into the abyss},
>   author={Owhadi, Houman and Yoo, Gene Ryan},
>   journal={Journal of Computational Physics},
>   volume={389},
>   pages={22--47},
>   year={2019},
>   publisher={Elsevier}
> }
>
> Due to weaknesses in novelty, lack of appropriate contextualization to prior work, and theoretical analysis that does not provide new insight, I must maintain my score.

---

### Official Review · Reviewer_BR88 · 2025-11-01

**Soundness:** 2
**Presentation:** 3
**Contribution:** 2
**Rating:** 4
**Confidence:** 3

**Summary:**

The paper deals with Function Encoders framework. The authors propose two PCA-inspired training algorithms: Progressive Training, which sequentially builds basis functions, and Train-Then-Prune, which trains many bases jointly and prunes redundant ones via covariance analysis.

Theoretical contributions include finite-sample generalization bounds based on Rademacher complexity and PAC-Bayes analysis, showing how generalization depends on the number of bases $n$ and their supremum norms, dataset size, and regularization $\lambda$.
Empirical studies on polynomial regression and nonlinear dynamical systems demonstrate that FEs achieve comparable accuracy to overparameterized neural networks using significantly fewer basis functions.

**Strengths:**

The paper is well-written and simple, making technical ideas accessible.

**Weaknesses:**

The theoretical analysis in the paper is detached from the heuristics it proposes. Since the framework is primarily heuristic, its experimental evaluation becomes the key validation tool, yet the experiments are limited in scope and scale. The benchmarks are relatively simple (toy and low-dimensional physical systems). Lacks comparison with other kernel-learning or low-rank approximation methods.

**Questions:**

How Theorem 1 and Theorem 2 can guide us in choosing better basis functions? Should we regularize $R^2n$ or $Rn^{3/2}$ at the stage of learning basis functions?

---

> ### Author Response · Authors · 2025-11-20
>
> 1. The theoretical analysis in the paper is detached from the heuristics it proposes.
>
> The connection between Sections 4 and 5 is through the number of basis functions $n$, which controls the complexity terms in both Theorem 1 and Theorem 2. Prior work in Kakade (2008) assumes a fixed-size feature map while our bounds explicitly specify the bounds in terms of $n$ and apply to a finite learned feature map.
>
> 2. Lacks comparison with other kernel-learning or low-rank approximation methods.
>
> We address this point in more detail in response to Reviewer nAUD. We clarify that function encoders learn an explicit set of basis functions across multiple tasks, thereby constructing a compact, finite-dimensional representation of the function space without assuming a kernel a priori. In contrast, other kernel methods like Nyström, RFF, and Deep Kernel Learning generally operate within the RKHS of the base kernel. Moreover, low-rank kernel approximations cannot recover this data-dependent RKHS because they only approximate a fixed $k$, and do not construct new basis functions. We will add additional information to the related work comparing with existing kernel learning and low-rank kernel approximation methods.
>
> 3. How Theorem 1 and Theorem 2 can guide us in choosing better basis functions?
>
> Theorems 1 and 2 determine that the bounds scale directly with the number of basis functions $n$, which aligns with well-known results on function space complexity. However, they do not prescribe which neural architecture should parameterize $\psi_j$, since architecture choice affects the approximation error, not the estimation error captured by our bounds. This separation is typical: theoretical bounds control risk within a fixed hypothesis class $F_n$, while architecture selection determines how well $F_n$ can approximate the true function class. Existing results in Ingebrand (2025) show that we can learn the underlying basis functions for the subspace spanning the data through the representation capacity of deep neural architectures, but in general, the question of which neural architecture to use to model a given dataset is an unsolved problem that is outside the scope of the current work.
>
> 4. Should we regularize $R^2n$ or $Rn^{3/2}$ at the stage of learning basis functions?
>
> Both quantities arise after training, once the basis functions are fixed, as artifacts of uniform-complexity bounds. During training of $\psi_j$, these norms are inaccessible because the geometry of the span is still evolving. For this reason, basis learning is regularized through standard network regularization (weight decay) while the $R^2 n$ or $Rn^{3/2}$ scaling pertains solely to the inference-stage analysis.

---

### Author Response · Authors · 2025-12-04

We thank the reviewers for the constructive comments. Reviewers found our approach promising but requested clearer exposition, better justification of novelty, and improved theoretical framing.

We have made the following major changes in our revision incorporating their feedback:

1. __Connection Between Algorithms and Theory (§4 → §5).__

We made the relationship between the CEV-based procedures and Theorems 1–2 explicit. Section 5 now states clearly that the train-then-prune procedure yields a feature map with effective rank $n_\tau$; once this basis is fixed, our generalization bounds apply directly with $n = n_\tau$. This highlights that the CEV criterion determines the model complexity appearing in Theorems 1–2.

2. __Clarification of N and Scope of Algorithm 1.__

We now state in the main experimental section that all benchmarks use $N = 1000$ training functions.
We also added text explaining that Algorithm 1 is intended for the regime $N \gg 1$, and that the special case N=1 reduces to the standard single-task encoder.

3. __Distinguishing Function Encoders from Standard Neural Networks.__

Section 3 now explicitly contrasts the two paradigms: standard neural networks learn a single predictor, whereas function encoders learn a shared basis and solve a task-specific linear problem within that basis. This clarifies why the encoder is not simply a multi-task neural net.

4. __Clarification of Novelty and the RKHS Viewpoint.__

Section 3 now emphasizes that function encoders learn the finite-dimensional RKHS itself—a shared subspace across tasks.
The induced kernel $k(x,x') = \langle \phi(x), \phi(x') \rangle$ is a consequence of learning $\phi$, not an object chosen a priori.
This contrasts explicitly with Nyström/RFF (which approximate a fixed RKHS) and with DKL (which adapts a kernel but retains O(m) inference complexity).

5. __Regularization Statement Corrected.__

We removed the sentence suggesting that increasing $\lambda$ necessarily improves generalization.

6. __Clarifications in §5.2 (PAC-Bayes).__

We clarified that Rademacher bounds are distribution-free, while PAC-Bayes yields tighter bounds only when the prior is data-dependent.
We also clarified why truncated Gaussians are used and how this extends classical PAC-Bayes analyses to unbounded losses.

7. __Correction in Introduction Regarding Kernel Adaptation.__

The introduction now states that classical kernel pipelines adapt hyperparameters inside a fixed RKHS; they do not learn the RKHS span itself. Our method instead learns the span and performs inference in the primal, independent of m.

8. __Clarification of $f_j$, $\hat f_j$, and Coefficients $c_j$.__

We expanded the explanation around Eq. 4:
* $f_j$ is the ground-truth function,
* $\hat{f}_j \in F_n$ is its encoder approximation,
* coefficients $c_j$ are not trained with the network but computed in closed form by ridge regression (Eq. 3).

---

### Note · Program_Chairs · 2026-01-06
**Submission Desk Rejected by Program Chairs**

The updated revision contains author names, violating double-blind.